# Long-term potentiation in neurogliaform interneurons modulates excitation–inhibition balance in the temporoammonic pathway

Marion S. Mercier, Vincent Magloire, Jonathan H. Cornford and Dimitri M. Kullmann [iD]

UCL Queen Square Institute of Neurology, Department of Clinical and Experimental Epilepsy, University College London, London, UK

Handling Editors: David Wyllie & Tommas Ellender

The peer review history is available in the Supporting information section of this article (https://doi.org/10.1113/JP282753#support-information-section).

**Abstract** Apical dendrites of pyramidal neurons integrate information from higher-order cortex and thalamus, and gate signalling and plasticity at proximal synapses. In the hippocampus, neuro-gliaform cells and other interneurons located within stratum lacunosum-moleculare (SLM) mediate powerful inhibition of CA1 pyramidal neuron distal dendrites. Is the recruitment of such inhibition itself subject to use-dependent plasticity, and if so, what induction rules apply? Here we show that interneurons in mouse SLM exhibit Hebbian NMDA receptor-dependent long-term potentiation

This article was first published as a preprint: Mercier MS, Magloire V, Cornford JH & Kullmann DM (2021). Long-term potentiation in neurogliaform interneurons modulates excitation-inhibition balance in the temporoammonic pathway. bioRxiv doi: 10.1101/531822

(LTP). Such plasticity can be induced by selective optogenetic stimulation of afferents in the temporoammonic pathway from the entorhinal cortex (EC), but not by equivalent stimulation of afferents from the thalamic nucleus reuniens. We further show that theta-burst patterns of afferent firing induces LTP in neurogliaform interneurons identified using neuron-derived neurotrophic factor (Ndnf)-Cre mice. Theta-burst activity of EC afferents led to an increase in disynaptic feed-forward inhibition, but not monosynaptic excitation, of CA1 pyramidal neurons. Activity-dependent synaptic plasticity in SLM interneurons thus alters the excitation–inhibition balance at EC inputs to the apical dendrites of pyramidal neurons, implying a dynamic role for these interneurons in gating CA1 dendritic computations.

(Received 16 December 2021; accepted after revision 19 July 2022; first published online 25 July 2022)
**Corresponding author** D. M. Kullmann: Queen Square, London WC1N 3BG, UK. Email: d.kullmann@ucl.ac.uk

**Abstract figure legend** Hebbian long-term potentiation (LTP) of excitatory transmission onto interneurons located within hippocampal stratum lacunosum-moleculare (SLM) can be induced by electrical stimulation protocols involving pairing of pre- and postsynaptic activity. Using Ndnf-Cre mice, we show that hippocampal neurogliaform (NGF) cells express this form of LTP. These cells receive glutamatergic afferents from both the nucleus reuniens of the thalamus and the entorhinal cortex (EC), but selective optogenetic activation of either set of fibres reveals LTP at EC inputs only. Using an optogenetic theta-burst stimulation (OptoTBS) protocol to stimulate EC fibres in a physiologically relevant way, we show that NGF interneuron LTP translates to an increase in disynaptic inhibition onto CA1 pyramidal cell distal dendrites. Monosynaptic EC–CA1 pyramidal cell inputs do not undergo equivalent potentiation, leading to a net decrease in the excitation/inhibition (E/I) ratio of this pathway.

## Key points

- Electrogenic phenomena in distal dendrites of principal neurons in the hippocampus have a major role in gating synaptic plasticity at afferent synapses on proximal dendrites. Apical dendrites also receive powerful feed-forward inhibition, mediated in large part by neurogliaform neurons.
- Here we show that theta-burst activity in afferents from the entorhinal cortex (EC) induces 'Hebbian' long-term potentiation (LTP) at excitatory synapses recruiting these GABAergic cells.
- LTP in interneurons innervating apical dendrites increases disynaptic inhibition of principal neurons, thus shifting the excitation–inhibition balance in the temporoammonic (TA) pathway in favour of inhibition, with implications for computations and learning rules in proximal dendrites.

## Introduction

Regenerative potentials in the apical dendrites of pyramidal neurons have a profound effect on plateau potentials, spiking and synaptic plasticity in more proximal dendritic segments (Basu et al., 2013; Dudman et al., 2007; Han & Heinemann, 2013; Jarsky et al., 2005; Takahashi & Magee, 2009), and have been implicated in the formation of new CA1 place fields (Bittner et al., 2015). Computational studies have suggested that interactions between distal and proximal dendrites can support the multiplexing of information, and provide a plausible implementation of the backpropagation algorithm for supervised learning (Guerguiev et al., 2017; Körding & König, 2001; Lillicrap et al., 2020; Naud & Sprekeler, 2018;

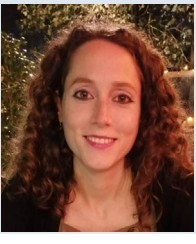

**Marion S. Mercier** is a senior postdoctoral research fellow at the UCL Queen Square Institute of Neurology. She obtained her PhD from Bristol University. She is an ARENA Associate Fellow, a mentor for the In2Science UK programme, and a founding member of the non-profit preprint peer-review initiative Peer Community In Circuit Neuroscience.

Payeur et al., 2021; Richards & Lillicrap, 2019; Sacramento et al., 2018; Urbanczik & Senn, 2014).

Importantly, the ability of excitatory afferents to distal dendrites to elicit regenerative events depends on the strength of disynaptic inhibition. In the hippocampus, the temporoammonic (TA) pathway from the entorhinal cortex (EC) powerfully recruits feedforward inhibition of apical dendrites in stratum lacunosum-moleculare (SLM). Indeed, bursts of TA activity can block both CA1 pyramidal cell spiking (Dvorak-Carbone & Schuman, 1999; Remondes & Schuman, 2002) and induction of long-term plasticity (Levy et al., 1998; Remondes & Schuman, 2002) in response to Schaffer collateral stimulation, effects that depend on GABAergic transmission. Given the importance of feedforward inhibition of distal dendrites, it is pertinent to ask whether excitatory synapses that recruit such interneurons are plastic, and if so, whether they exhibit 'Hebbian' NMDA-receptor-dependent long-term potentiation (LTP) as reported in stratum radiatum interneurons (Lamsa et al., 2005, 2007a), rather than other forms of plasticity reported in parvalbumin-positive interneurons in stratum oriens (Lamsa et al., 2007b; Oren et al., 2009; Perez et al., 2001). Apical dendrites in CA1 also receive a direct innervation from the higher order thalamic nucleus reuniens (NRe) (Dolleman-van der Weel et al., 2019), which also recruits feedforward inhibition (Dolleman-Van der Weel & Witter, 2000; Dolleman-Van der Weel et al., 1997). A further question, therefore, is whether plasticity rules differ between synapses made by cortical and subcortical afferents.

While a heterogeneous population of interneurons resides within SLM, or on the border between SLM and stratum radiatum (Capogna, 2011; Klausberger, 2009; Vida et al., 1998), the neurogliaform (NGF) subtype is the most abundant (Bezaire & Soltesz, 2013), and likely has a dominant role in SLM feedforward inhibition. NGF cells are innervated both by TA afferents and by axons from NRe (Chittajallu et al., 2017). They signal via volume transmission (Oláh et al., 2009), evoking exceptionally long-lasting inhibitory responses mediated by both $GABA_A$ and $GABA_B$ receptors (Price et al., 2008), and closely mimicking the effect of electrical stimulation of SLM (Williams & Lacaille, 1992). These postsynaptic response kinetics make NGF cells particularly well suited to regulating dendritic non-linearities and burst firing in pyramidal cells (Schulz et al., 2018). Interest in the properties and roles of NGF cells (Overstreet-Wadiche & McBain, 2015) has accelerated with the creation of a mouse line selective for neocortical layer 1 NGF cells, neuron-derived neurotrophic factor (Ndnf)-Cre (Tasic et al., 2016), although this has not, to date, been characterised extensively in the hippocampus.

Here, we first identify and characterise LTP mechanisms in SLM feedforward interneurons, and show, using an optogenetic strategy, that LTP and short-term facilitation can be induced at excitatory inputs from the EC but not at those from the thalamus. We confirm that the Ndnf-Cre mouse line can be used to target hippocampal NGF cells, and use it to demonstrate that this inhibitory cell subtype exhibits LTP in response to a physiologically relevant pattern of afferent activity. We further show that plasticity of feed-forward inhibition from the EC is not matched by parallel potentiation at EC–CA1 distal dendrite pyramidal cell synapses, and therefore that it alters the excitation–inhibition balance in the TA pathway.

## Methods

### Ethical approval

All procedures were performed in accordance with the UK Home Office Animals (Scientific Procedures) Act 1986 under license: PPL-PAF2788F5. All mice were housed under a non-reversed 12 h:12 h light–dark cycle and had access to food and water *ad libitum*.

### Animals

Hippocampal slices were prepared from male and female wild-type mice (C57BL/6; Harlan, Bicester, UK) that were postnatal day 14–25 for electrically induced LTP experiments, or 2–5 months old for optogenetic experiments. Heterozygous Ndnf-Cre breeding pairs were obtained from The Jackson Laboratory (B6.Cg-Ndnf[tm1.1(folA/cre)Hze]/J; Stock No: 028536; Bar Harbor, ME, USA), and bred on a C57BL/6 background. Male and female 3- to 5-month-old Ndnf-Cre mice were used for NGF cell LTP experiments.

### Surgery for viral injections

Mice (minimum age 6 weeks) were anaesthetised using isoflurane and given metacam (0.1 mg/kg) and buprenorphine (0.02 mg/kg) subcutaneously for pain relief post-surgery. They were then positioned in a stereotaxic frame and placed on a heated pad to maintain body temperature. Bilateral craniotomies were made for hippocampal and EC injections, or a single hole was made in the midline for injections into the NRe. The coordinates used for each location were: EC, −4.3 mm caudal and ±3.0 mm lateral of Bregma, and −3.2 and −3.0 mm deep from the pia with the syringe angled at 12−14°; NRe, −0.7 mm caudal, on the midline, and −4.4, −4.2 and −4.0 mm deep; dorsal CA1 (SLM): −2.0 mm caudal and ±1.25 mm lateral, −1.65 and −1.45 mm deep. The viral suspension (200 nl) was injected at each site (100 nl/min) through a 33 gauge needle using a

Hamilton syringe, and the needle was left for 5 min post-injection before being withdrawn. The viruses used were AAV5-CaMKIIα-hchR2(H134R)-mCherry-WPRE, AAV5-EF1α-DIO-EYFP and AAV5-EF1a-DIO-hChR2-mCherry (all titres $>10^{12}$ viral genomes/ml) and were purchased from UNC Vector Core (Chapel Hill, NC, USA). Animals were given 0.5 ml saline subcutaneously to help with recovery and monitored for ∼5 days following the procedure. Acute experiments were performed a minimum of 3 weeks post-surgery to allow for viral expression.

## Slice preparation

Young mice (P14–25) were decapitated under isoflurane anaesthesia and the brains were rapidly removed and placed in ice-cold, sucrose-rich slicing solution composed of (in mM): sucrose (75), NaCl (87), KCl (2.5), $NaH_2PO_4$ (1.25), $NaHCO_3$ (25), glucose (25), $CaCl_2$ (0.5), $MgCl_2$ (7), and oxygenated with 95% $O_2$–5% $CO_2$. Older mice (≥2 months) were given a lethal dose of sodium pentobarbital, and trans-cardially perfused with ice-cold oxygenated sucrose solution, before removal of the brain. Slices (300 $\mu$m-thick) were cut in a sagittal or horizontal orientation; results from experiments on sagittal and horizontal slices did not differ and were pooled for analysis. Slices were cut in ice-cold sucrose solution using a vibrating blade-microtome (Leica VT1200 S; Leica Microsystems, Wetzlar, Germany), and were left to recover for 15 min at 32°C before being transferred to a standard artificial CSF (aCSF) solution containing (in mM): NaCl (119), KCl (2.5), $MgSO_4$ (1.3), $NaH_2PO_4$ (1.25), $NaHCO_3$ (25), glucose (11), $CaCl_2$ (2.5), and oxygenated with 95% $O_2$–5% $CO_2$. Slices were allowed to recover for at least 1 h at room temperature before being transferred to a recording chamber for experiments. The CA3 region was removed from slices in experiments where GABA receptors were blocked to prevent recurrent activity.

## Electrophysiology

**Recording.** Slices were held in place in a submerged recording chamber continuously perfused with oxygenated aCSF at a rate of 2–3 ml/min and maintained at 30–32°C. Picrotoxin (100 $\mu$M) and CGP55845 (1 $\mu$M) were routinely added to block $GABA_A$ and $GABA_B$ receptors, respectively, unless stated otherwise. Slices were visualised using an upright microscope (BX51WI, Olympus, Tokyo, Japan) and differential interference contrast (DIC) optics were used to identify cells located in the SLM layer of CA1. In wild-type mice, cells with small (∼10 $\mu$m), round somata were selected for experiments, in line with confirmed NGF

cell morphology (Price et al., 2005). In Ndnf-Cre mice, EYFP- or mCherry-tagged neuron-derived neurotrophic factor (NDNF)+ cells were identified using blue (470 nm) or orange (590 nm) LED illumination, respectively (ThorLabs, Newton, NJ, USA), delivered through the microscope objective (×20; Olympus). Whole cell patch clamp recordings were performed using borosilicate glass micropipettes (2–4 M$\Omega$) containing, for current clamp experiments, a $K^+$-based internal solution containing (in mM): potassium gluconate (125), KCl (5), KOH–Hepes (10), $MgCl_2$ (1), sodium phosphocreatine (10), EGTA (0.2), Mg-ATP (4), $Na_3$-GTP (0.4), and for voltage clamp experiments a $Cs^+$-based solution containing (in mM): caesium gluconate (125), NaCl (8), CsOH–Hepes (10), EGTA (0.2), Mg-ATP (4), sodium phosphocreatine (10), $Na_3$-GTP (0.3), TEA-Cl (5). Solutions were adjusted to pH 7.2–7.5 and 290−295 mOsm, and biocytin (0.4%) was added for *post hoc* morphological analysis in a subset of experiments. The liquid junction potentials for the potassium gluconate and caesium gluconate based internal solutions were 15.7 and 17.4 mV, respectively. Recordings were made using a Multiclamp 700B amplifier (Molecular Devices, San Jose, CA, USA), filtered at 10 kHz and digitised at 20 kHz, and experiments were run using custom LabVIEW (National Instruments, Austin, TX, USA) virtual instruments. Recordings were not corrected for liquid junction potentials, and cells were rejected if they had an access resistance $>25$ M$\Omega$, or access or input resistance that changed by more than 20% over the course of the experiment. Recordings for the low frequency stimulation pairing (LFS-P), some of the theta burst stimulation (TBS) (Figs 4 and 7*A*), and the spike-pairing experiments were made in current clamp mode at −70 mV, except during the LFS-P induction protocol when cells were temporarily switched to voltage clamp and depolarised to −10 mV. Disynaptic inhibitory post-synaptic currents (IPSCs) and monosynaptic excitatory postsynaptic currents (EPSCs) in pyramidal cells were recorded in voltage clamp at +10 mV and −70 mV respectively. For LTP of monosynaptic EPSCs, cells were temporarily switched to current clamp for the duration of the TBS protocol.

**Stimulation.** Concentric bipolar stimulating electrodes (FHC, Bowdoin, ME, USA) were positioned in the SLM and stimulation was delivered at 0.05 Hz via constant current stimulators (Digitimer, Welwyn Garden City, UK). In experiments where two pathways were tested, stimulating electrodes were positioned on either side of the recorded cell, at opposite ends of SLM, and pathway independence was verified by comparing responses to paired-pulse stimulation within and across pathways: paired pulses with an inter-pulse interval of 50 ms were applied to each pathway and typically revealed

paired-pulse facilitation (PPF) ((excitatory postsynaptic potential (EPSP)$_1$ slope/EPSP$_2$ slope) $\times$ 100; mean $\pm$ SD: $176 \pm 38\%$; $n = 58$). Paired pulses were then applied across pathways, and pathways were considered independent if responses were not facilitated following stimulation of the other pathway. It should be noted that spikes were often triggered by larger EPSPs, thus preventing the measurement of PPF and pathway overlap in some cells. During experiments, stimulation was delivered to each pathway alternately, or to a single pathway, at a frequency of 0.05 Hz. The stimulation intensity for all electrically induced LTP experiments was set to elicit sub-maximal EPSPs measuring $\sim$5 mV in amplitude. For optogenetic stimulation of thalamic or EC axons, 1 ms-long pulses of blue light (470 nm) were delivered through the microscope objective to elicit sub-maximal EPSPs of amplitude $\geq$2.5 mV, or EPSCs or disynaptic IPSCs of $\geq$50 pA. Optogenetic stimulation for these experiments was delivered at 0.03 Hz to prevent use-dependent depression of NGF cell responses over time (Price et al., 2005).

Electrical or optogenetic LTP induction protocols were as follows: LFS-P: 200 pulses delivered at 2 Hz, while holding the cell at $-10$ mV in voltage clamp (for dual pathway experiments, no stimulation was applied to the control pathway during this time); spike pairing: 30 pulses delivered at 0.2 Hz, paired with 5 ms-long depolarising current steps to induce one AP in the post-synaptic cell $\sim$10 ms after presynaptic stimulation (cells were maintained at $-65$ mV in current clamp between current steps throughout this protocol to ensure sufficient depolarisation for AP firing); TBS: 10 bursts of five pulses at 100 Hz delivered at theta frequency (5 Hz), and repeated 3 times, 30 s apart, while the postsynaptic membrane potential was allowed to float.

Electrophysiological properties of cells were assessed by injection of 1 s-long current steps, ranging from $-200$ pA to $+300$ pA in 50 pA increments. Rheobase current was also identified (minimal current injection required to evoke 1 AP), and a 1 s-long current step at twice rheobase delivered in order to assess firing frequency and adaptation; the latter was measured as the ratio between the firing frequency observed during the last 200 ms and that seen during the first 200 ms of the current step. Input resistance was measured and averaged from the $-50$ and $+50$ pA steps, and membrane time constant ($\tau$) was calculated by fitting a single exponential function to the first 150 ms of the $-50$ pA step. Sag was calculated in two ways from the $-200$ pA current step: (1) as an absolute measure (mV) of the negative deflection seen at the start of the step, taken as the difference between the peak negative value and the steady state voltage measured at the end of the step (last 300 ms), and (2) this absolute value expressed as a percentage of the total negative deflection evoked by the $-200$ pA current injection,

to take into account differences in input resistance across cells. AP and afterhyperpolarisation (AHP) properties were calculated from the single AP evoked at rheobase current.

## Cell fixation and histochemistry

Cells were filled with biocytin during whole-cell recordings and slices were placed in 4% paraformaldehyde (PFA) overnight at 4°C. Fixed slices were rinsed three times in phosphate buffered saline (PBS), and sub-sequently stored in PBS containing 0.02% sodium azide. Biocytin was revealed by incubating slices in PBS with 0.3% Triton X-100 and streptavidin-conjugated Alexa Fluor (488 or 594, 1:1000; Thermo Fisher Scientific, Waltham, MA, USA, 21832 or S11227) for 3 h at room temperature. Some of the slices from Ndnf-Cre mice that had been injected with AAV5-EF1$\alpha$-DIO-EYFP were also stained for green fluorescent protein (GFP) to enhance the enhanced yellow fluorescent protein (EYFP) tag and confirm the identity of recorded cells. For these stainings, slices were first incubated in blocking solution consisting of 0.5% Triton X-100, 0.5% BSA and 10% goat serum in PBS for 1 h at room temperature, before being transferred to PBS containing 0.5% Triton X-100, 0.5% BSA and guinea pig anti-GFP antibody (1:1000; Synaptic Systems, Goettingen, Germany, 132 005) and left overnight at 4°C. After washing in PBS, slices were incubated in goat anti-guinea pig secondary antibody conjugated with Alexa Fluor 488 (1:500; Thermo Fisher Scientific, A-11073) and streptavidin-conjugated Alexa Fluor 594 (1:1000; Thermo Fisher Scientific, S1127). 4′,6-Diamidino-2-phenylindole (DAPI) staining was then performed by incubating slices for 5 min in PBS with DAPI (1:5000).

For characterisation of reelin expression, Ndnf-Cre mice injected with AAV5-EF1$\alpha$-DIO-EYFP were trans-cardially perfused with 4% PFA and brains were removed and post-fixed in the same solution for at least 24 h. After rinsing with PBS, brains were sliced coronally (70 $\mu$M) and reelin staining was performed as described above, using mouse anti-reelin (1:1000; Abcam, Cambridge, UK, ab78540) and goat anti-mouse Alexa Fluor 568 (1:500; Thermo Fisher Scientific A-11004). Expression was quantified in three hippocampal sections from three injected mice ($n = 9$): the SLM was manually selected as the region of interest in each slice and reelin+ and NDNF+ EYFP cells were counted separately before over-laying images to quantify overlap.

## Statistical analysis

For all LTP experiments, $n$-values represent cells from separate slices, obtained from at least three different

mice per experiment. All experiments involving pharmacological manipulations were interleaved with control experiments. Individual experiments were analysed using custom code written in Python, and statistical analysis was carried out in Python and Origin (2018; OriginLab Corp., Northampton, MA, USA). For quantification of LTP, a paired Student's *t*-test was performed on raw pre- (baseline) *versus* post-LTP (last 10 min) EPSP slope or amplitude measurements. EPSP slope was measured during a 2 ms window from the onset of the EPSP. Test and control pathways were compared using paired *t*-tests on normalised responses averaged from the last 10 min of experiments, and unpaired *t*-tests were used to compare pharmacological manipulations and optogenetic responses to thalamic *versus* EC stimulation. The Shapiro–Wilk test was used to assess normality, and paired and unpaired *t*-tests were replaced by Wilcoxon's signed rank test and the Mann–Whitney *U*-test, respectively, if data were found to be non-normally distributed. Spearman's rank correlation was used to assess correlations. Differences were considered significant when $P < 0.05$, and are indicated in the figures as $P < 0.001$ (***), $P < 0.01$ (**), $P < 0.05$ (*), or $P = x$ when $0.05 < P < 0.1$. All time-series data are presented as means ± SEM and all summary statistics are presented as means ± SD. Representative traces are an average of 15 traces taken during the 5 min baseline period or during the last 5 min of recordings, unless stated otherwise.

## Results

### LTP in SLM interneurons

We recorded from interneurons in CA1 SLM with small round somata, consistent with NGF cell morphology (Price et al., 2005), and stimulated afferent fibres in the same layer (Fig. 1*A*). We restricted attention to the initial slope of EPSPs and applied a low-frequency stimulation (LFS) protocol paired with postsynaptic depolarisation (LFS-P; Fig. 1*A*) to look for LTP at monosynaptic excitatory inputs (Maccaferri & McBain, 1996). Inhibitory transmission was blocked by inclusion of GABA$_A$ and GABA$_B$ receptor antagonists in the aCSF. LFS-P led to an increase in EPSP slope to 137 ± 42% of baseline, lasting for at least 30 min (Fig. 1*B*; $n = 22$; $t(21) = -4.01$, $P = 0.0006$, paired *t*-test). EPSP slopes after LFS-P ranged from 77% to 211% of baseline (Fig. 1*B*). While insufficient morphological reconstructions were achieved to conclusively identify all cell types, a number of electrophysiological properties consistent with an NGF cell profile correlated positively with LTP magnitude: namely, a larger input resistance in the range of 200 MΩ ($n = 22$, $\rho = 0.52$, $P = 0.013$, Spearman correlation; Fig. 1*C*), a relatively depolarised action potential threshold ($n = 22$, $\rho = 0.54$, $P = 0.010$, Spearman correlation; Fig. 1*D*), and a large AHP ($n = 22$, $\rho = 0.42$, $P = 0.050$, Spearman correlation; Fig. 1*E*) (Price et al., 2005; Tricoire et al., 2010). Action potential latency did not correlate significantly with LTP magnitude ($n = 22$, $\rho = 0.40$,

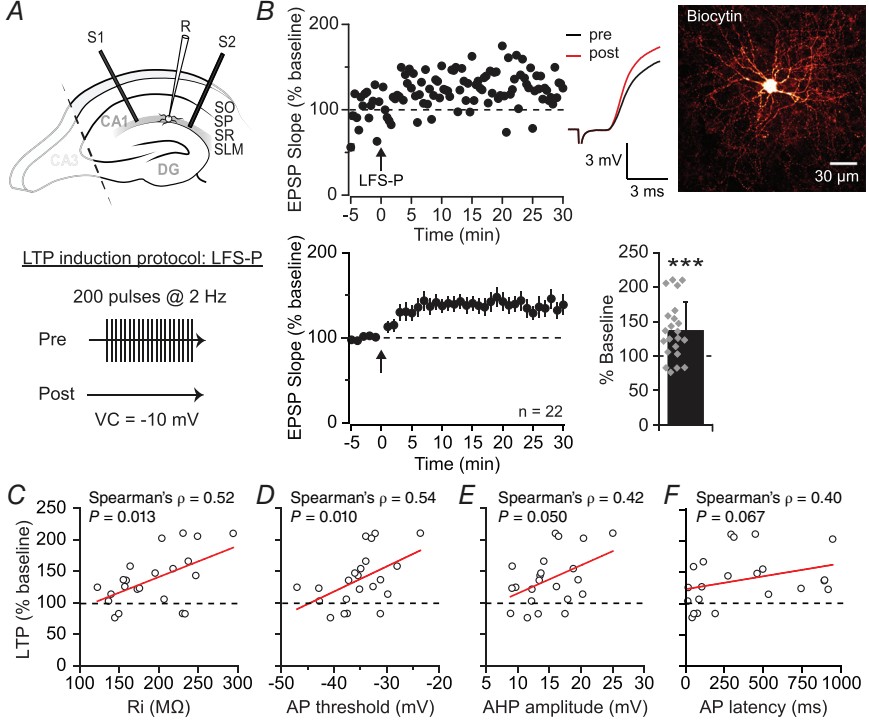

**Figure 1. Pairing-induced LTP in SLM interneurons**
*A*, top, experimental set-up showing placement of one recording (R) and two stimulating (S1, S2) electrodes in SLM of a hippocampal slice with CA3 removed. Bottom, schematic representation of the low-frequency stimulation-pairing (LFS-P) LTP induction protocol. VC, voltage clamp. *B*, representative experiment and traces (top left) in a biocytin-filled NGF-like cell (top right) and pooled dataset (bottom) showing LTP in SLM interneurons (quantified bottom right; $P = 0.0006$). Arrows indicate LTP induction protocol. *C–F*, scatter plots illustrating the relationship between input resistance ($R_i$; *C*), action potential threshold (AP; *D*), afterhyperpolarisation amplitude (AHP; *E*) or AP latency (*F*) and LTP magnitude. The individual cells are the same as in *B* ($n = 22$). [Colour figure can be viewed at wileyonlinelibrary.com]

$P = 0.067$, Spearman correlation; Fig. 1F), which may be due to a tendency for NGF cells to fire either early or late during threshold current injections (Jiang et al., 2015).

Dual pathway experiments (schematic diagram: Fig. 1A) revealed a significantly larger increase in the paired pathway than in a control pathway that was not stimulated during the LFS-P protocol (test: $152 \pm 43\%$, control: $124 \pm 27\%$, $n = 10$; $t(9) = 2.88$, $P = 0.018$, paired *t*-test; Fig. 2A), implying a degree of pathway specificity. The small yet significant potentiation observed in the control pathway ($t(9) = -3.20$, $P = 0.011$, paired *t*-test) could potentially be due to incomplete independence of the two stimulated pathways. However, on average there was no facilitation of responses in the control pathway when stimulated after the test pathway ($102 \pm 24\%$), in contrast to robust PPF seen when applying paired pulses to the control pathway alone ($197 \pm 28\%$; $n = 6$, $t(5) = 6.33$, $P < 0.0001$, paired *t*-test; Fig. 2B, see also Methods), suggesting significant pathway separation. We therefore tentatively conclude that LTP is not fully restricted to stimulated synapses, as might be expected in cells with aspiny dendrites (Chen & Sabatini, 2012; Cowan et al., 1998), such as SLM interneurons (Lacaille & Schwartzkroin, 1988) and NGF cells (Price et al., 2005).

Consistent with LTP in principal cells that depends on NMDA receptors, LFS-P-induced LTP in SLM interneurons was fully blocked by perfusion of DL-aminophosphonovalerate (APV; 100 $\mu$M; interleaved

LFS-P: $142 \pm 16\%$, $n = 5$; APV: $95 \pm 18\%$, $n = 5$; $t(8) = 4.49$, $P = 0.002$, unpaired *t*-test; Fig. 2C). Importantly, in control experiments neither presynaptic LFS alone nor postsynaptic depolarisation alone led to potentiation, further confirming its Hebbian nature (LFS alone: $96 \pm 37\%$, $n = 6$; $t(5) = 0.23$, $P = 0.831$, paired *t*-test, Fig. 2D; depolarisation alone: $111 \pm 19\%$, $n = 5$; $t(4) = -1.32$, $P = 0.257$, paired *t*-test, Fig. 2E).

In contrast to LFS-P, a spike pairing protocol consisting of 30 presynaptic stimuli, each followed by a single postsynaptic spike (Fig. 3A) induced a potentiation that resisted NMDA receptor blockade (spike pairing: $136 \pm 32\%$, $n = 15$; $t(14) = -4.55$, $P = 0.0005$, paired *t*-test; APV: $135 \pm 27\%$, $n = 6$; $t(5) = 3.55$, $P = 0.016$, paired *t*-test; with and without APV: $t(19) = 0.09$, $P = 0.932$, unpaired *t*-test; Fig. 3B). Instead, this was blocked by the voltage-gated $Ca^{2+}$ channel blockers nimodipine (10 $\mu$M) and $Ni^{2+}$ (100 $\mu$M) ($109 \pm 29\%$, $n = 5$; $t(4) = -0.39$, $P = 0.715$, paired *t*-test; Fig. 3C), indicating that NMDA receptor-independent $Ca^{2+}$ influx, likely triggered by back-propagating action potentials, can also be sufficient for LTP induction in these cells, as previously reported in other hippocampal interneurons (Galván et al., 2008; Nicholson & Kullmann, 2017). Some interaction with dendritic EPSPs is likely, however, as eliciting 30 postsynaptic spikes alone induced a non-sustained increase in EPSP slope ($123 \pm 31\%$, $n = 8$; $t(7) = -2.08$, $P = 0.076$, paired *t*-test; Fig. 3D).

**Figure 2. Pairing-induced LTP is Hebbian and NMDA receptor-dependent**

*A*, LFS-P-induced LTP was relatively pathway-specific, quantified on right ($P = 0.018$). *B*, pathway independence was verified by comparing the paired-pulse ratio (PPR) across and within pathways ($P < 0.0001$). Inset shows example traces in response to stimulation of the control (grey) or test (black) pathways; responses indicated by the thick lines should be equivalent if pathways are fully independent. *C*, perfusion of APV (100 $\mu$M) blocked the induction of LTP by LFS-P ($P = 0.002$). *D*, pooled dataset and representative traces showing no LTP in SLM interneurons following presynaptic LFS only ($P = 0.831$; scale bars: 2.5 mV, 5 ms). *E*, as for *D* but for postsynaptic depolarisation only ($P = 0.257$; scale bars: 4 mV, 5 ms). [Colour figure can be viewed at wileyonlinelibrary.com]

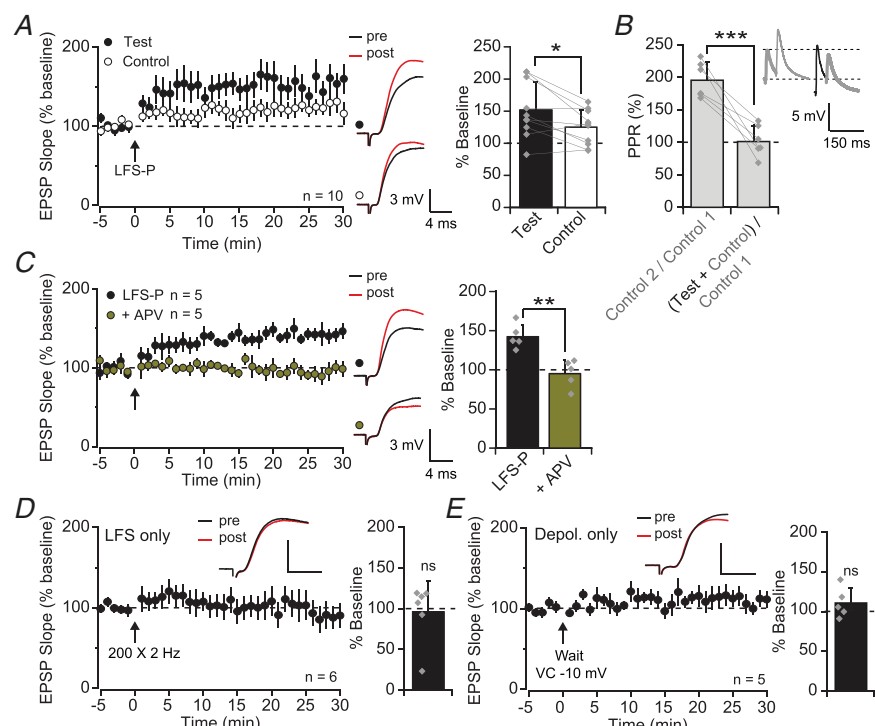

## LTP induction by stimulation of EC but not thalamic inputs to SLM interneurons

The SLM region of CA1 receives excitatory projections from both layer III of the EC (Witter et al., 1988) and NRe of the thalamus (Wouterlood et al., 1990). Electrical stimulation in SLM, however, does not distinguish between afferents from these regions. In order to activate either pathway selectively, we injected an adeno-associated virus (AAV) encoding channelrhodopsin-2 (ChR2) under the CaMKII promoter into either the NRe or the EC (Fig. 4A–C), and recorded responses in SLM interneurons to optogenetic stimulation of ChR2-expressing axons (Fig. 4B and C). Importantly, viral injections in either location led to expression of ChR2 within the hippocampus that was restricted to SLM (Fig. 4B and C), and, in the case of EC injections, the molecular layer of the dentate gyrus (Fig. 4B), as shown previously (Chittajallu et al., 2017). Light stimulation reliably evoked EPSPs in SLM interneurons following expression of ChR2 in either set of axons (Fig. 4B and C). Although optogenetic responses evoked by stimulation of EC afferents were stable (97 ± 17% after 30 min recording, $n = 13$; $t(12) = 0.64$, $P = 0.536$, paired $t$-test; Fig. 4D), those evoked by stimulation of thalamic afferents gradually increased with time (137 ± 29% after 30 min recording, $n = 6$; $t(5) = -2.22$, $P = 0.077$; Fig. 4E). The mechanisms underlying this 'run-up' were not investigated, but may be related to progressive $Ca^{2+}$ loading of presynaptic terminals following repetitive ChR2 activation (Zhang & Oertner, 2007). The instability of optogenetically evoked thalamic EPSPs complicates the ability to measure LTP in this pathway. We therefore used light stimulation exclusively to deliver a conditioning stimulus selectively to either EC or thalamic terminals, and used electrical stimulation to monitor EPSPs at synapses made by both sets of afferents (Fig. 4F and G). Pairing optogenetic stimulation of EC inputs with depolarisation, while electrical stimulation was interrupted, led to an increase in EPSP slope to 138 ± 47% of baseline ($n = 12$; $t(11) = 3.85$, $P = 0.003$, paired $t$-test; Fig. 4F). In contrast, the same pairing applied to thalamic afferents had no effect (EPSP slope 107 ± 30% of baseline, $n = 11$; $t(10) = -0.44$, $P = 0.671$, paired $t$-test; Fig. 4G). Notably, light-evoked responses were of similar amplitude whether stimulating EC or thalamic terminals, both during baseline recordings of EPSPs in current clamp (EC: 5.7 ± 2.0 mV, $n = 12$; thalamus: 6.6 ± 2.6 mV, $n = 11$; $t(21) = -0.92$, $P = 0.370$, unpaired $t$-test; Fig. 4H) and during the LTP induction protocol in voltage clamp (EC: 59.0 ± 31 pA, $n = 12$; thalamus: 50.2 ± 21 pA, $n = 11$; $U = 58$, $P = 0.644$, Mann–Whitney $U$-test; Fig. 4I), suggesting that the different outcomes could not be explained by the degree of SLM interneuron depolarisation. Furthermore, a TBS protocol (see below) applied to thalamic fibres also failed to induce a significant potentiation of EPSPs (119 ± 26%, $n = 8$; $t(7) = -1.76$, $P = 0.122$, paired $t$-test; Fig. 4J). Interestingly, optogenetically evoked responses to stimulation of EC afferents, but not thalamic axons, exhibited PPF (EC: 156 ± 60%, $n = 14$; thalamus: 99 ± 26%, $n = 13$; $t(25) = 3.10$, $P = 0.005$, unpaired $t$-test; Fig. 4K), consistent with a lower release probability in the TA pathway. The differences between these pathways potentially contribute to the variability in LTP magnitude seen with electrical induction (Fig. 1) if the ratio of EC and NRe afferents varies depending on the position of the stimulating electrode.

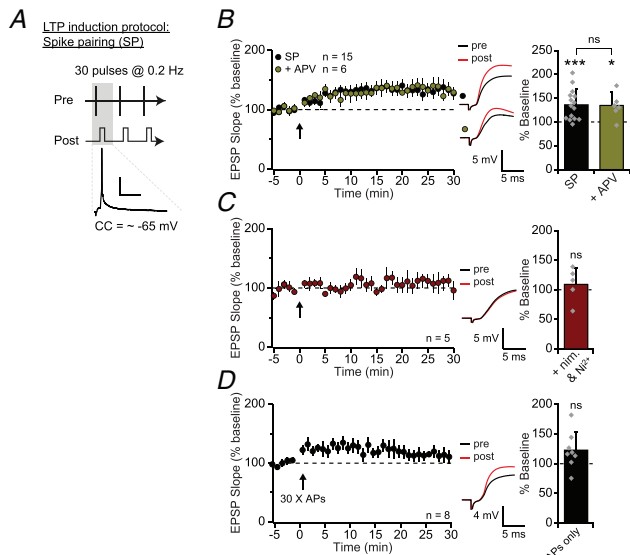

**Figure 3. Spike-pairing evoked LTP is resistant to NMDA blockade**
*A*, schematic representation of the spike pairing protocol (below: example trace from a representative postsynaptic cell; scale bars: 25 mV, 40 ms). CC, current clamp. *B*, pooled dataset and representative traces showing that spike pairing-induced LTP was not prevented by perfusion of APV, quantified on right (SP: $P = 0.0005$; +APV: $P = 0.016$; group comparison: $P = 0.932$). *C*, the spike-pairing protocol failed to induce LTP in the presence of $Ni^{2+}$ (100 μM) and nimodipine (10 μM; nim) ($P = 0.715$). *D*, postsynaptic action potentials alone failed to induce a persistent increase in EPSP slope ($P = 0.076$). [Colour figure can be viewed at wileyonlinelibrary.com]

## Hippocampal NDNF+ cells correspond to NGF cells

NGF cells constitute ~10% of the total inhibitory neuron population in hippocampal area CA1, and the overwhelming majority are located in SLM (Bezaire & Soltesz, 2013). The neurons recorded from in the present study had a somatic shape typical of NGF cells. Indeed, biocytin filling and *post hoc* morphological reconstruction confirmed that at least some of the cells in which LTP was studied were indeed NGF interneurons (Fig. 1B).

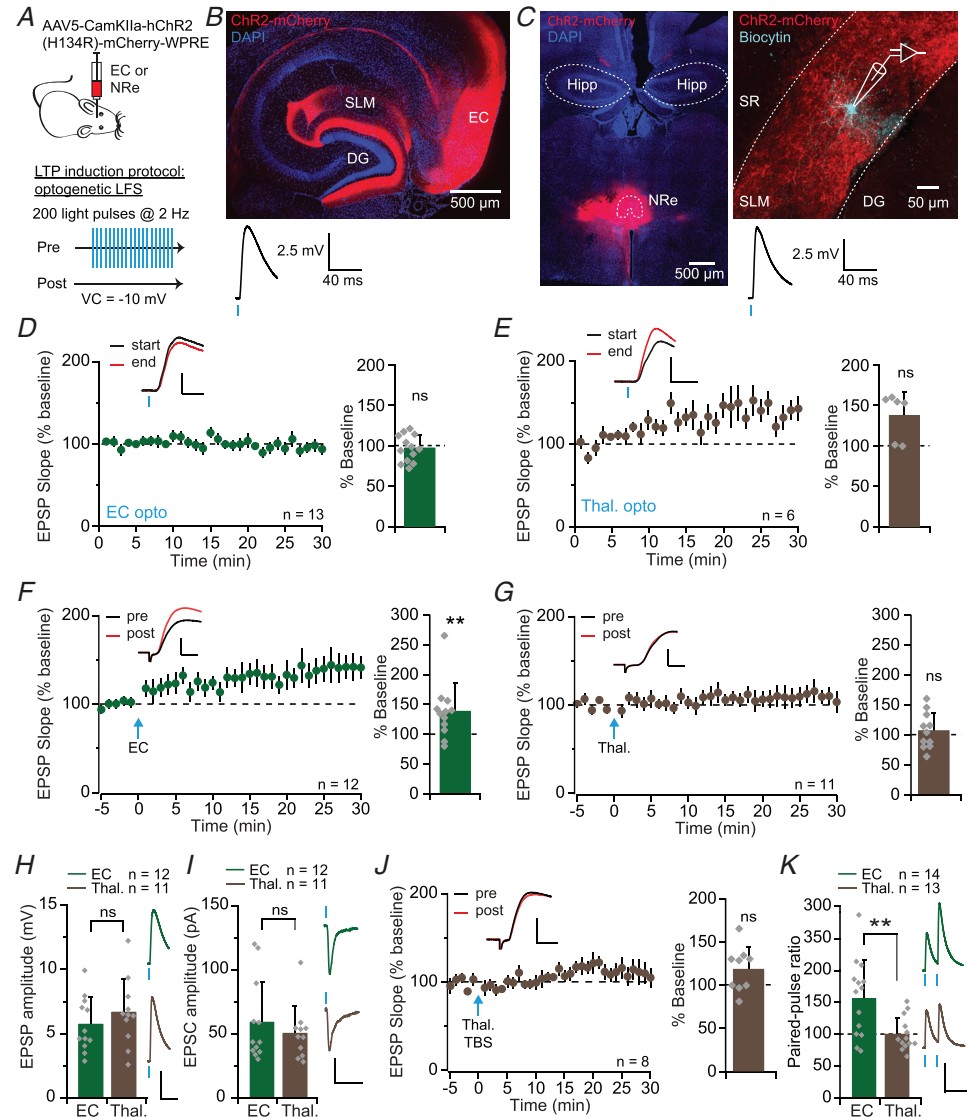

**Figure 4. LTP induction by stimulation of entorhinal cortex but not thalamic inputs to SLM interneurons**
*A*, top: schematic diagram showing viral strategy used to target the nucleus reuniens (NRe) of the thalamus or the entorhinal cortex (EC); bottom: schematic diagram of the optogenetic low-frequency stimulation-pairing (LFS-P) protocol. *B*, confocal image of a horizontal brain section (70 μm) showing viral expression (red) at the injection site in the EC, as well as in perforant path axons in the SLM region of hippocampal CA1 and molecular layer of the dentate gyrus (DG), and DAPI staining (blue). Below: example light-induced EPSP recorded from an SLM interneuron in a mouse injected in the EC (average 15 traces). *C*, confocal images of a coronal brain section (70 μm) showing viral expression (red) at the injection site in the thalamus including the NRe, and DAPI staining (blue; left), and a sagittal hippocampal slice (300 μm) showing a biocytin-filled SLM interneuron (cyan) surrounded by transduced thalamic afferents (red; right). Below: example light-induced EPSP recorded from the same cell (average 15 traces). DG, dentate gyrus; Hipp., hippocampus, SR, stratum radiatum. *D*, optogenetic stimulation of EC afferents evoked EPSPs in SLM interneurons that remained stable over a 30 min recording period, quantified on right (*P* = 0.536; scale bars: 1 mV, 5 ms). *E*, as for *D* but showing run-up of optogenetically evoked thalamic responses in SLM interneurons (*P* = 0.077; scale bars: 3 mV, 4 ms). *F*, optogenetic LFS-P of EC afferents induced LTP, quantified on right (*P* = 0.003; scale bars: 3 mV, 4 ms). *G*, the same optogenetic stimulation protocol applied to thalamic afferents did not induce LTP (*P* = 0.671; scale bars: 3 mV, 4 ms). *H*, optogenetic stimulation of EC or thalamic afferents evoked EPSPs of similar amplitude, recorded during the baseline period of LTP experiments (average of 15 traces; *P* = 0.370; scale bars: 2.5 mV, 40 ms). *I*, EPSCs recorded during the optogenetic LFS-P LTP induction protocol (average of 200 traces) were of similar amplitude for both sets of afferents (*P* = 0.644; scale bars: 25 pA, 40 ms). *J*, a theta-burst stimulation (TBS) protocol applied to thalamic afferents also failed to induce LTP (*P* = 0.122; scale bars: 2.5 mV, 5 ms). *K*, paired-pulse facilitation (PPF) was induced by optogenetic paired-pulse stimulation of EC but not thalamic axons (right: averages of five traces) (*P* = 0.0005; scale bars: 5 mV, 100 ms). [Colour figure can be viewed at wileyonlinelibrary.com]

However, their fine dendritic and axonal arborisations render morphological recovery particularly challenging (Tremblay et al., 2016). In order to directly address whether hippocampal NGF cells express LTP, we used Ndnf-Cre mice, a mouse line shown to specifically target NGF cells in cortical layer I (Tasic et al., 2016; see also Abs et al., 2018; Schuman et al., 2019). As NDNF has principally been identified as an NGF cell-specific marker in the neocortex, we first sought to characterise hippocampal expression of NDNF+ cells by injecting the dorsal hippocampus of Ndnf-Cre mice with an AAV encoding either Cre-dependent EYFP or Cre-dependent ChR2 tagged with mCherry (Fig. 5*A*). NDNF+ cells

**Table 1. Electrophysiological properties of hippocampal NDNF+ cells**

| Property | Value |
|---|---|
| Resting potential (mV) | $-62.1 \pm 2.1$ |
| Input resistance (MΩ) | $221.5 \pm 17.4$ |
| Time constant ($\tau$) (ms) | $12.0 \pm 0.5$ |
| Sag (mV) | $1.8 \pm 0.6$ |
| Sag (%) | $5.1 \pm 0.8$ |
| AP amplitude (mV) | $47.8 \pm 1.8$ |
| AP half-width (ms) | $1.03 \pm 0.05$ |
| AP latency (ms) | $399.6 \pm 86.8$ |
| AP threshold (mV) | $-32.6 \pm 1.5$ |
| AHP amplitude (mV) | $14.8 \pm 0.9$ |
| Firing frequency (Hz) | $28.0 \pm 3.1$ |
| Adaptation ratio | $0.6 \pm 0.1$ |

Data are means $\pm$ SEM ($n = 20$).

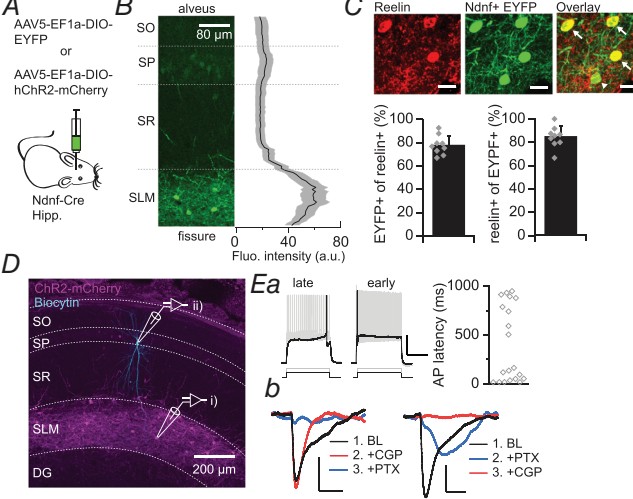

**Figure 5. Hippocampal NDNF+ cells are NGF cells**
*A*, schematic diagram showing viral strategy used to express EYFP or ChR2-mCherry in hippocampal NDNF+ cells. Hipp., hippocampus. *B*, left, confocal image of a section of CA1 used to measure NDNF+-EYFP fluorescence intensity across hippocampal layers. Right, mean (black) $\pm$ SEM (grey) fluorescence intensity across hippocampal layers ($n = 6$ slices, four mice), normalised to percentage distance from the alveus to the hippocampal fissure. SO, stratum oriens; SP, stratum pyramidale; SR, stratum radiatum. *C*, close-up of the SLM showing reelin immunostaining (top left), EYFP+ NDNF cells (top centre) and overlay (top right; scale bar: 20 $\mu$m). White arrows indicate reelin-positive NDNF cells and arrowhead shows one reelin-negative NDNF cell. Percentage overlap was quantified in nine slices from three mice (bottom). *D*, confocal image of a sagittal hippocampal slice (300 $\mu$m) showing selective expression in SLM neurons (magenta), and a biocytin-filled pyramidal cell (cyan) in stratum pyramidale. DG, dentate gyrus. *E*, electrophysiological properties of NDNF+ cells and projections to pyramidal neurons. *Ea*, mCherry-tagged NDNF+ cells showed late- or early-spiking properties in response to a near-rheobase current step (black; quantified on right, $n = 20$), and sustained spiking in response to a double-rheobase current step (grey; scale bars: 25 mV, 500 ms). *Eb*, optogenetic stimulation of NDNF+ cells induced long-lasting inhibitory postsynaptic potentials (IPSPs) in pyramidal cells at baseline (BL) (black) that were abolished by blockers of GABA$_A$ (blue; PTX: picrotoxin, 100 $\mu$M) and GABA$_B$ (red; CGP: CGP 55845, 1 $\mu$M) receptors (scale bars: 1 mV, 125 ms). [Colour figure can be viewed at wileyonlinelibrary.com]

were abundant in the hippocampus, and were found almost exclusively in SLM (Fig. 5*B* and *D*; fluorescence intensity analysed in six slices from four mice), consistent with the known location of NGF cells. Weak expression was occasionally observed in both CA1 and dentate gyrus cell body layers, but never in hippocampal strata where other interneuron subtypes are located, such as radiatum and oriens; this suggests a possible minor leak in excitatory cells, but likely selective targeting of NGF cells amongst other hippocampal interneuron subtypes. Indeed, immunostaining for reelin, a molecular marker selectively expressed by hippocampal NGF cells (Fuentealba et al., 2010), confirmed that the overwhelming majority of NDNF+ cells in the SLM were reelin-positive (84.5 $\pm$ 9%, $n = 9$ slices from 3 mice; Fig. 5*C*). Furthermore, threshold current injections in NDNF+ cells revealed two populations with either early- or late-spiking properties ($n = 20$; Fig. 5*Ea*), matching the firing pattern described in both cortical NDNF+ (Tasic et al., 2016) and NGF (Jiang et al., 2015) cells; other electrophysiological properties were also within the ranges described for both hippocampal (Price et al., 2005; Tricoire et al., 2010) and cortical NGF (Jiang et al., 2015; Karagiannis et al., 2009) cells, as well as NDNF+ cells (Tasic et al., 2016) (Table 1). Importantly, these properties were also in line with those found here to correlate positively with LTP magnitude (Fig. 1*C*–*F*). Finally, optogenetic activation of hippocampal NDNF+ cells produced long-lasting inhibitory responses in pyramidal neurons comprising a GABA$_{A\text{-slow}}$ and a strong GABA$_B$ component, evident both in the biphasic nature of the response and in the pharmacological dissection of its components (Fig. 5*Eb* and Table 2), both of which are hallmarks of NGF cell signalling (Capogna & Pearce, 2011; Price et al., 2005; Tamas, 2003). Together, these properties

**Table 2. Properties of IPSPs in pyramidal cells evoked by optogenetic stimulation of NDNF+ cells**

| Property | Baseline (n = 7) | GABA_A* (n = 4) |
|---|---|---|
| Rise time (ms) | 29.9 ± 2.5 | 25.3 ± 2.5 |
| Decay (τ) (ms)† | — | 44.9 ± 3.0 |
| Amplitude (mV) | −3.5 ± 0.7 | −5.0 ± 1.2§ |
| Half-width (ms) | 155.7 ± 31.1 | 72.4 ± 5.4 |

Data are p means ± SEM. In one cell, GABA_A receptors were first blocked with picrotoxin (100 μM), revealing a long-lasting GABA_B-mediated IPSP with amplitude −1.7 mV and half-width 239.7 ms (see Fig. 5*Eb*).
*Isolated by application of CGP 55845 (1 μM).
†Baseline responses were biphasic.
§IPSP amplitude increased following application of CGP 55845, likely because of blockade of presynaptic GABA_B receptors (Price et al., 2008).

strongly support the use of Ndnf-Cre mice for the selective targeting of hippocampal NGF cells.

## Hippocampal NGF cells exhibit LTP

Having confirmed the identity of hippocampal NDNF+ cells, we next sought to verify that LTP could be induced in NGF cells by recording from fluorescently tagged NDNF+ cells in SLM and applying the LFS-P protocol with electrical stimulation (Fig. 6*A* and *B*). This induced an increase in EPSP slope to 144 ± 55% of baseline ($n = 8$; $Z = 2.45$, $P = 0.008$, Wilcoxon's signed rank test; Fig. 6*C*), confirming that hippocampal NGF cells exhibit robust LTP. We then asked whether this form of plasticity occurs under more physiological conditions by excluding GABA receptor blockers from the aCSF to leave inhibitory transmission intact, and by applying a TBS protocol to mimic activity at EC–CA1 synapses (Buzsáki, 2002) (Fig. 6*D*). Under these conditions, excitatory transmission onto NGF cells was significantly potentiated to 138 ± 30% of baseline ($n = 8$; $Z = −2.45$, $P = 0.014$, Wilcoxon's signed rank test; Fig. 6*E*), indicating that excitatory inputs onto hippocampal NGF cells are likely to be potentiated by firing patterns that occur in the TA pathway *in vivo*.

## SLM feedforward interneuron LTP alters the excitation–inhibition balance at EC–CA1 synapses

What effect does LTP at synapses on SLM interneuron have on the downstream hippocampal network? Specifically, how does LTP of disynaptic inhibition of CA1 pyramidal neurons interact with LTP of monosynaptic excitation? The TBS protocol is not only physiologically relevant but also avoids the need for experimentally imposed depolarisation of postsynaptic

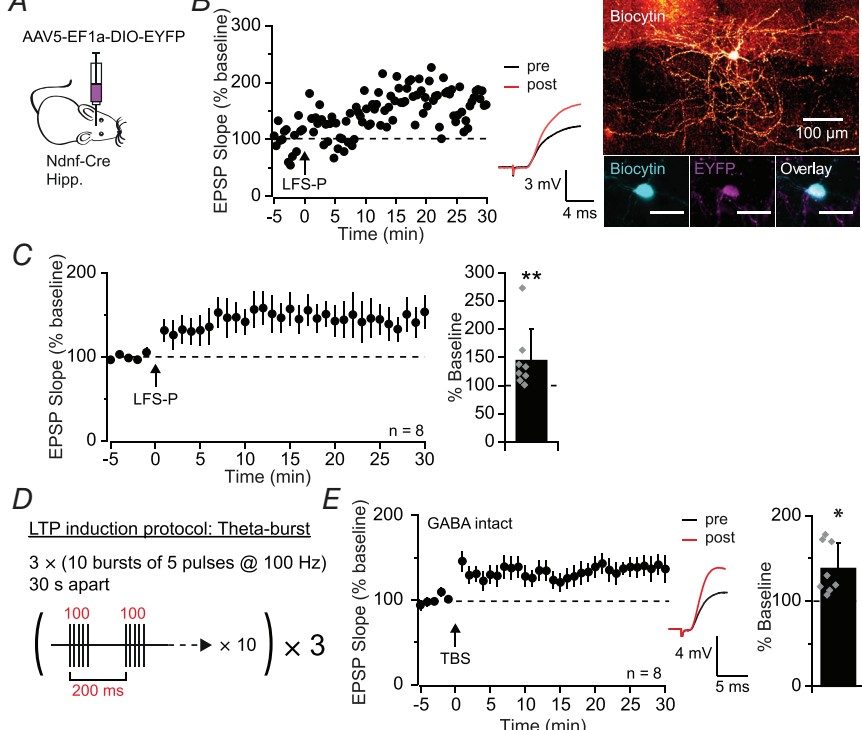

**Figure 6. Low-frequency and theta-burst stimulation induced LTP in hippocampal NGF cells**
*A*, schematic diagram showing viral strategy used to express EYFP in hippocampal NDNF+ cells. Hipp., hippocampus. *B*, representative experiment and traces of LFS-P-induced LTP (left) in a biocytin-filled EYFP-tagged NDNF+ cell (right; bottom right scale bars: 30 μm). *C*, pooled dataset showing LFS-P-induced LTP in NDNF+ NGF cells, quantified on right (*P* = 0.008). *D*, schematic representation of the theta-burst stimulation (TBS) LTP induction protocol. *E*, pooled dataset showing TBS-induced LTP with GABAergic transmission intact in NDNF+ NGF cells (*P* = 0.014). [Colour figure can be viewed at wileyonlinelibrary.com]

cells via a patch clamp pipette, and should therefore allow LTP to be induced in a population of interneurons mediating disynaptic inhibition. However, a potential obstacle to recording disynaptic inhibition in CA1 pyramidal cells is that electrical stimulation in SLM also recruits interneurons directly. Indeed, inhibitory postsynaptic currents (IPSCs) evoked by electrical stimulation in SLM and recorded in CA1 pyramidal cells held in voltage clamp at +10 mV were not abolished by blocking AMPA receptors with 2,3-Dioxo-6-nitro-1,2,3,4-tetrahydrobenzo[f]quinoxaline-7-sulfonamide (NBQX; 10 $\mu$M) (Fig. 7*A*). IPSCs persisted in the presence of NBQX even when the stimulating electrode was placed as far from the recorded pyramidal cell as possible (103 ± 16 %, *n* = 4; *t*(3) = −0.23, *P* = 0.834, paired *t*-test; Fig. 7*B*). This finding is in line with a previous study showing that electrical stimulation of the SLM as far as 900 $\mu$m from the recorded pyramidal cell directly recruits GABAergic neurons mediating monosynaptic inhibition (Milstein et al., 2015), and is likely due to the high density of interneurons with profuse axonal arborisations located within SLM (Armstrong et al., 2012a). We therefore switched to an optogenetic strategy to elicit disynaptic inhibition of CA1 pyramidal neurons: IPSCs evoked by light pulses following ChR2 expression in the EC were abolished by NBQX (10 ± 16 %, *n* = 4; *t*(3) = 3.13, *P* = 0.050, paired *t*-test; Fig. 7*B*). When optogenetic stimulation of EC axons was used both to elicit disynaptic inhibition and to deliver the TBS protocol, LTP was reliably induced, detected as an increase in feedforward inhibition (135 ± 27%, *n* = 10; *t*(9) = −4.10, *P* = 0.003, paired *t*-test; Fig. 7*C* and *D*).

Finally, we asked whether this form of LTP serves to maintain the excitation–inhibition (E–I) balance of the EC input in the face of potentiation occurring in parallel at excitatory inputs onto CA1 pyramidal cells, as has been shown for LTP of feedforward inhibition in stratum radiatum (Lamsa et al., 2005). In order to record monosynaptic EPSCs (similarly elicited by optogenetic EC stimulation), we held pyramidal cells at −70 mV, near the GABA$_A$ reversal potential, in voltage clamp, but switched to current clamp during the LTP induction protocol to allow the postsynaptic membrane potential to depolarise naturally in response to the optogenetic TBS. Under these conditions, designed to match those that induce LTP of feedforward inhibition, we saw no potentiation of EPSCs in CA1 pyramidal cells (99 ± 14%, *n* = 10; *t*(9) = −0.02, *P* = 0.982, paired *t*-test; monosynaptic EPSCs *versus* disynaptic IPSCs: *t*(18) = 3.76, *P* = 0.001, unpaired *t*-test; Fig. 7*D*). Taken together, these results suggest that LTP at synapses on SLM interneurons alters the E–I balance in the EC input to hippocampal CA1 in favour of inhibition.

## Discussion

The present study shows that interneurons located in SLM express Hebbian, NMDA receptor-dependent LTP, and that this can be induced by optogenetic stimulation of inputs originating in EC layer III but not, under the conditions tested here, by stimulation of afferents from the NRe of the thalamus. Importantly, we confirm that a mouse line previously developed to target cortical NGF cells, Ndnf-Cre (Tasic et al., 2016), also selectively

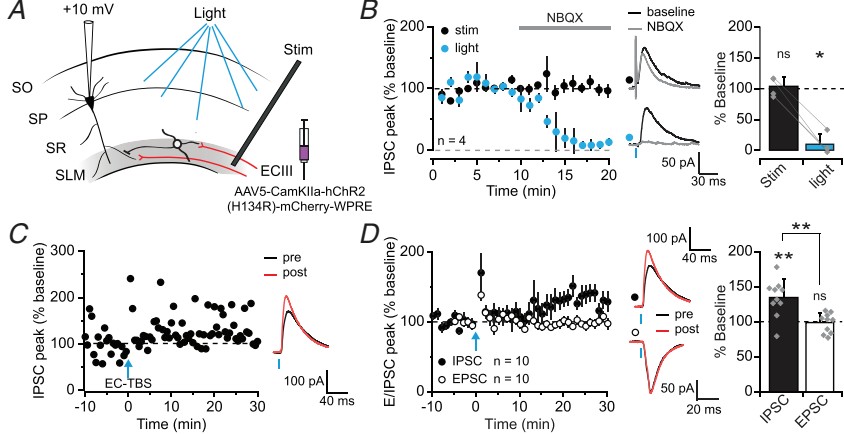

**Figure 7. LTP of SLM interneurons alters excitation–inhibition balance in the temporoammonic pathway**
*A*, experimental set-up for recording disynaptic IPSCs from CA1 pyramidal cells elicited by electrical stimulation of the SLM (stim) or optogenetic stimulation of EC afferents (light). SO, stratum oriens; SP, stratum pyramidale; SR, stratum radiatum. *B*, effect of NBQX (10 $\mu$M) on IPSCs elicited by alternate electrical and optogenetic stimulation. Reduction of IPSCs by NBQX quantified on right (stim: *P* = 0.834; light: *P* = 0.050). *C*, representative experiment and traces showing LTP of feedforward inhibition induced by optogenetic TBS of EC afferents (EC-TBS). *D*, pooled dataset showing EC-TBS-induced LTP of disynaptic IPSCs (black; *P* = 0.002), but not of monosynaptic EPSCs (white; *P* = 0.982), with representative traces (IPSCs same as in panel *D*) (group comparison: *P* = 0.001). [Colour figure can be viewed at wileyonlinelibrary.com]

labels hippocampal NGF cells, and show, using this novel tool, that this prominent interneuron subtype expresses LTP. Finally, using a theta-burst protocol resembling the natural activity of EC inputs to CA1, with inhibition intact, we show that plasticity is readily induced at EC–SLM interneuron synapses but not at monosynaptic connections from EC to pyramidal cell distal dendrites, and that this translates into a downstream shift in E–I balance in favour of inhibition.

The finding that CA1 interneurons located in SLM, of which NGF cells account for a large fraction, exhibit long-term plasticity has not, to our knowledge, previously been shown. Indeed, one earlier study reported no evidence of LTP, although it focused on interneurons located at the border of SLM and stratum radiatum with stimulation of excitatory fibres in both strata (Ouardouz & Lacaille, 1995). The present findings, however, complement work showing plasticity at mossy fibre inputs to SLM interneurons located in CA3 (Galván et al., 2008), as well as evidence of LTP in CA1 stratum radiatum Ivy cells (Szabo et al., 2012), a cell-type closely related to NGF cells (Armstrong et al., 2012b; Tricoire et al., 2010), and of plasticity reported in cortical NDNF+ NGF cells (Abs et al., 2018). We found that LTP in SLM interneurons is typically NMDA receptor-dependent and at least partially pathway-specific. This is in line with previous reports of pathway-specific, NMDA receptor-dependent Hebbian LTP in aspiny hippocampal interneurons (Galván et al., 2015; Lamsa et al., 2005), and further challenges the view that spines are necessary for the dendritic compartmentalisation of plasticity (Nimchinsky et al., 2002; Yuste et al., 2000). Interestingly, we found that a spike pairing protocol induced an NMDA receptor-independent form of plasticity in these cells, which was instead dependent on $Ca^{2+}$ influx through VGCCs. This phenomenon is reminiscent of plasticity mechanisms seen in oriens-alveus interneurons, where NMDA receptor-independent LTP requires $Ca^{2+}$-permeable AMPA receptors, T-type $Ca^{2+}$ channels or nicotinic receptors (Nicholson & Kullmann, 2014, 2017, 2021). Together, these results argue that postsynaptic $Ca^{2+}$ is necessary for LTP induction in SLM interneurons, but that the source of $Ca^{2+}$ itself can vary.

Notably, selective optogenetic stimulation of EC afferents induced LTP in SLM interneurons whilst equivalent stimulation of thalamic afferents did not. This could be due to a number of pre- and/or post-synaptic differences between these two synapses, such as the specific receptors present, the probability of release at each synapse, and the resultant short-term plasticity mechanisms that may be at play during the induction protocol. In support of the latter hypothesis, optogenetic dissection of the two inputs revealed robust paired-pulse facilitation only at synapses made by afferents from the EC. Although short-term facilitation has previously

been reported in response to NRe stimulation *in vivo* (Bertram & Zhang, 1999; Dolleman-Van der Weel et al., 1997, 2017; Gruart et al., 2015), these studies did not examine synapses on inhibitory cells. It is possible that NRe synapses on SLM interneurons exhibit a higher basal release probability than those onto pyramidal cells, thereby resulting in an absence of paired-pulse facilitation at the former. Indeed, previous studies have shown that neurotransmitter exocytosis probability can vary in a target cell-dependent manner, even across synapses formed by the same presynaptic axon (Branco & Staras, 2009; Koester & Johnston, 2005).

In the broader context of feedback connections and their computational function, the different short- and long-term plasticity rules revealed here at cortical and subcortical inputs from the EC and thalamus, respectively, are particularly interesting. Indeed, the plasticity displayed by EC inputs onto SLM interneurons corresponds to recent computational work proposing that hierarchically connected networks can coordinate learning via burst-dependent plasticity and multiplexing of top-down and bottom-up signals (Payeur et al., 2021). In this model, short-term facilitation is required for signal multiplexing, while LTP of inhibition may regulate pyramidal cell plasticity via control of burst probability. In contrast, the absence of either short- or long-term plasticity at NRe inputs, which are also thought to be part of a higher-order cortico-thalamo-cortical circuit (Dolleman-van der Weel et al., 2019), suggests that alternative mechanisms may be involved in coordinating learning between these structures. Alternatively, neuro-modulation, for instance mediated by local cholinergic inputs, may be required to reveal plasticity at NRe–SLM interneuron synapses, as has been suggested for stratum oriens interneurons (Nicholson & Kullmann, 2021). These open questions warrant further experimental and computational investigation.

Ndnf-Cre mice have, hitherto, primarily been used to label NGF cells located in cortical layer I (Abs et al., 2018; Tasic et al., 2016). Our results, however, indicate that this mouse line also enables selective targeting of NGF cells within the hippocampus, in line with previously reported expression of NDNF in this sub-cortical structure (Kuang et al., 2010). Thus, we show that hippocampal NDNF+ cells are located primarily within SLM, express the NGF cell-associated molecular marker reelin (Fuentealba et al., 2010), exhibit electro-physiological properties that are consistent with those previously described for both hippocampal (Tricoire et al., 2010) and cortical (Jiang et al., 2015; Karagiannis et al., 2009; Tasic et al., 2016) NGF cells, and induce slow, long-lasting $GABA_A$ and $GABA_B$ receptor-mediated post-synaptic responses typical of NGF interneuron signalling (Capogna & Pearce, 2011; Oláh et al., 2009; Price et al., 2005; Tamas, 2003). While these results strongly

support selectivity for NGF cells amongst hippocampal interneurons, weaker expression was also occasionally observed in putative excitatory cells. Hippocampal NGF cells have previously been difficult to target or manipulate selectively. Indeed, whilst neuronal nitric oxide synthase (Bloss et al., 2016; Li et al., 2014; Taniguchi et al., 2011) and neuropeptide Y (Chittajallu et al., 2013; Jackson et al., 2018; Krook-Magnuson et al., 2011; Li et al., 2017; Tricoire et al., 2010), amongst others, have been used as markers to aid in their identification, neither is fully selective for NGF cells (Overstreet-Wadiche & McBain, 2015; Pelkey et al., 2017). The Ndnf-Cre mouse line is thus the first genetic tool to achieve such selectivity, and will no doubt prove invaluable for investigations of hippocampal NGF cell function. Furthermore, a recent study showing that NDNF is a conserved marker for NGF cells in the human cortex (Poorthuis et al., 2018) implies that results obtained with this mouse line could be translated to humans.

The finding that TBS with intact GABAergic transmission, reminiscent of natural activity patterns and conditions found in the direct EC–CA1 pathway (Buzsáki, 2002), induces LTP in SLM interneurons and NGF cells, but not in pyramidal cells, is notable. On a mechanistic level, failure to elicit LTP of the excitatory TA pathway may be due to the distal location of EC inputs on pyramidal cell dendrites, which, when combined with powerful feedforward inhibition, limits the ability of the TA pathway to drive membrane depolarisation in these cells. Importantly, the downstream effect is a net shift in E–I balance in favour of inhibition, in contrast to the effect of LTP in feedforward stratum radiatum interneurons reported previously (Lamsa et al., 2005). Indeed, LTP in these inhibitory cells was shown to counterbalance potentiation occurring in parallel at Schaffer collateral–CA1 pyramidal cell synapses, and thereby to maintain E–I balance and preserve the temporal fidelity of synaptic integration. Feedforward inhibition in the SLM is thought to impose a time window within which TA and Schaffer collateral inputs can interact non-linearly (Capogna, 2011; Overstreet-Wadiche & McBain, 2015), and thus likely has a restrictive role on the generation of dendritic spikes (Jarsky et al., 2005), plateau potentials and plasticity (Bittner et al., 2015; Remondes & Schuman, 2002) in CA1 pyramidal cells. Interestingly, inhibition in layer I of the cortex, mediated largely by NGF cells (Jiang et al., 2015), has been proposed to have an analogous role in regulating dendritic non-linearities generated by input coupling (Larkum et al., 1999), and activation of cortical NDNF+ NGF cells has been shown to inhibit the generation of dendritic spikes in layer V pyramidal neurons (Abs et al., 2018). It seems likely, then, that plasticity of excitatory inputs onto NGF cells, both within the hippocampus and in the neocortex (if indeed it occurs there), and the resultant shift in E–I balance, will lead to a tighter regulation of supra-linear dendritic integration in pyramidal cells. Future experiments studying this phenomenon directly, and employing *in vivo* optogenetic strategies in Ndnf-Cre mice, will be instrumental in understanding the impact of potentiating TA feedforward inhibition on the wider hippocampal network and behaviour.

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

## Additional information

### Data availability statement

The raw data that support the findings of this study will be made available from the corresponding author upon request.

### Competing interests

The authors declare no competing financial interests.

### Author contributions

M.S.M., V.M., J.H.C. and D.M.K. completed the work at University College London. M.S.M., V.M., J.H.C. and D.M.K. conceived and designed the experiments. M.S.M. performed the experiments and analysed the data, with additional experiments and analysis by V.M. and J.H.C. All authors interpreted the results of the experiments. M.S.M. and D.M.K. drafted the manuscript, which was edited and approved by all authors, who agree to be accountable for all aspects of the work in ensuring that questions related to the accuracy or integrity of any part of the work are appropriately investigated and resolved. All persons designated as authors qualify for authorship, and all those who qualify for authorship are listed.

### Funding

This work was supported by the Wellcome Trust, the Medical Research Council and Epilepsy Research UK.

### Author's present address

J. H. Cornford: Mila, Montreal, QC, H2S 3H1, Canada.

### Keywords

hippocampus, interneurons, long-term potentiation, neurogliaform cells

## Supporting information

Additional supporting information can be found online in the Supporting Information section at the end of the HTML view of the article. Supporting information files available:

**Statistical Summary Document**
**Peer Review History**

