## [Peer Review History · The Journal of Physiology]

Long-term potentiation in neurogliaform interneurons modulates excitation-inhibition balance in the temporoammonic pathway

Marion Mercier, Vincent Magloire, Jonathan Cornford, and Dimitri M Kullmann

DOI: 10.1113/JP282753

Corresponding author(s): Dimitri Kullmann (d.kullmann@ucl.ac.uk)

The following individual(s) involved in review of this submission have agreed to reveal their identity: Cheng-Chang Lien (Referee #1)

Review Timeline:

Submission Date:	16-Dec-2021
Editorial Decision:	17-Jan-2022
Revision Received:	20-Jun-2022
Accepted:	19-Jul-2022

Senior Editor: David Wyllie

Reviewing Editor: Tommas Ellender

Transaction Report:

Dear Professor Kullmann,

Re: JP-RP-2021-282753 "Long-term potentiation in stratum lacunosum-moleculare interneurons modulates excitation-inhibition balance in the temporoammonic pathway" by Marion Mercier, Vincent Magloire, Jonathan Cornford, and Dimitri M Kullmann

Thank you for submitting your manuscript to The Journal of Physiology. It has been assessed by a Reviewing Editor and by 2 expert Referees and I am pleased to tell you that it is considered to be acceptable for publication following satisfactory revision.

The reports are copied at the end of this email. Please address all of the points and incorporate all requested revisions, or explain in your Response to Referees why a change has not been made.

NEW POLICY: In order to improve the transparency of its peer review process The Journal of Physiology publishes online as supporting information the peer review history of all articles accepted for publication. Readers will have access to decision letters, including all Editors' comments and referee reports, for each version of the manuscript and any author responses to peer review comments. Referees can decide whether or not they wish to be named on the peer review history document.

Authors are asked to use The Journal's premium BioRender (<https://biorender.com/>) account to create/redrawn their Abstract Figures. Information on how to access The Journal's premium BioRender account is here: <https://physoc.onlinelibrary.wiley.com/journal/14697793/biorender-access> and authors are expected to use this service. This will enable Authors to download high-resolution versions of their figures.

I hope you will find the comments helpful and have no difficulty returning your revisions within 4 weeks.

Your revised manuscript should be submitted online using the links in Author Tasks Link Not Available.

Any image files uploaded with the previous version are retained on the system. Please ensure you replace or remove all files that have been revised.

REVISION CHECKLIST:

- Article file, including any tables and figure legends, must be in an editable format (eg Word)
- Abstract figure file (see above)
- Statistical Summary Document
- Upload each figure as a separate high quality file
- Upload a full Response to Referees, including a response to any Senior and Reviewing Editor Comments;
- Upload a copy of the manuscript with the changes highlighted.

- A potential 'Cover Art' file for consideration as the Issue's cover image;
- Appropriate Supporting Information (Video, audio or data set https://jp.msubmit.net/cgi-bin/main.plex?form_type=display_requirements#supp).

To create your 'Response to Referees' copy all the reports, including any comments from the Senior and Reviewing Editors, into a Word, or similar, file and respond to each point in colour or CAPITALS and upload this when you submit your revision.

I look forward to receiving your revised submission.

If you have any queries please reply to this email and staff will be happy to assist.

Yours sincerely,

REQUIRED ITEMS:

-Author photo and profile. First (or joint first) authors are asked to provide a short biography (no more than 100 words for one author or 150 words in total for joint first authors) and a portrait photograph. These should be uploaded and clearly labelled with the revised version of the manuscript. See Information for Authors for further details.

-You must start the Methods section with a paragraph headed Ethical Approval. A detailed explanation of journal policy and regulations on animal experimentation is given in Principles and standards for reporting animal experiments in The Journal of Physiology and Experimental Physiology by David Grundy J Physiol, 593: 2547-2549. doi:10.1113/JP270818.). A checklist outlining these requirements and detailing the information that must be provided in the paper can be found at: <https://physoc.onlinelibrary.wiley.com/hub/animal-experiments>. Authors should confirm in their Methods section that their experiments were carried out according to the guidelines laid down by their institution's animal welfare committee, and conform to the principles and regulations as described in the Editorial by Grundy (2015). The Methods section must contain details of the anaesthetic regime: anaesthetic used, dose and route of administration and method of killing the experimental animals.

-Your manuscript must include a complete Additional Information section

-Please upload separate high-quality figure files via the submission form.

-A Statistical Summary Document, summarising the statistics presented in the manuscript, is required upon revision. It must be on the Journal's template, which can be downloaded from the link in the Statistical Summary Document section here: https://jp.msubmit.net/cgi-bin/main.plex?form_type=display_requirements#statistics

-Papers must comply with the Statistics Policy https://jp.msubmit.net/cgi-bin/main.plex?form_type=display_requirements#statistics

In summary:

-If n {less than or equal to} 30, all data points must be plotted in the figure in a way that reveals their range and distribution. A bar graph with data points overlaid, a box and whisker plot or a violin plot (preferably with data points included) are acceptable formats.

-If $n > 30$, then the entire raw dataset must be made available either as supporting information, or hosted on a not-for-profit repository e.g. FigShare, with access details provided in the manuscript.

-' n ' clearly defined (e.g. x cells from y slices in z animals) in the Methods. Authors should be mindful of pseudoreplication.

-All relevant ' n ' values must be clearly stated in the main text, figures and tables, and the Statistical Summary Document (required upon revision)

-The most appropriate summary statistic (e.g. mean or median and standard deviation) must be used. Standard Error of the Mean (SEM) alone is not permitted.

-Exact p values must be stated. Authors must not use 'greater than' or 'less than'. Exact p values must be stated to three significant figures even when 'no statistical significance' is claimed.

-Statistics Summary Document completed appropriately upon revision

-Please include an Abstract Figure. The Abstract Figure is a piece of artwork designed to give readers an immediate

understanding of the research and should summarise the main conclusions. If possible, the image should be easily 'readable' from left to right or top to bottom. It should show the physiological relevance of the manuscript so readers can assess the importance and content of its findings. Abstract Figures should not merely recapitulate other figures in the manuscript. Please try to keep the diagram as simple as possible and without superfluous information that may distract from the main conclusion(s). Abstract Figures must be provided by authors no later than the revised manuscript stage and should be uploaded as a separate file during online submission labelled as File Type 'Abstract Figure'. Please ensure that you include the figure legend in the main article file. All Abstract Figures should be created using BioRender. Authors should use The Journal's premium BioRender account to export high-resolution images. Details on how to use and access the premium account are included as part of this email.

EDITOR COMMENTS

Reviewing Editor:

Both Reviewers agree that these findings are exciting.

They have further suggestions for additional experiments and points for clarification ((e.g. is the ability of certain synapses to undergo plasticity dependent on the properties synapses or method of activation (electrical vs optical)).

In addition they suggest an increase in robustness of results is needed by increasing in 'n' for certain experiments. This might also affect the statistical analysis that is most appropriate (non-parametric or t-tests).

Senior Editor:

Your manuscript has been positively assessed by two expert referees and an Reviewing Editor. Notwithstanding their overall positive comments, they have raised concerns about some aspects of the work which will require further experimentation. Aside from the specific queries, they note that some experiments would appear to be under-powered and I would encourage you to ensure that that data you report are statistically robust and analysed appropriately. Please also ensure you report n values clearly - I note that in places you give x slices from y animals but in other places you simply report the number of cells (from how many slices, how many animals?). I look forward to receiving a revised manuscript in due course.

Please ensure that any revised manuscript complies with the Statistics Policy - for example, report SD (not SEM), precise P values (unless $P < 0.0001$)

REFEREE COMMENTS

Referee #1:

The authors of this manuscript investigated long-term potentiation (LTP) in interneurons in the stratum lacunosum-moleculare (SLM) and how it increases di-synaptic inhibition of pyramidal cells, thus finally shifting excitation-inhibition balance at the entorhinal cortex (EC) input to pyramidal cell synapses. They demonstrated that SLM interneurons, which were mostly identified as neurogliaform cells, expressed Hebbian NMDA receptor-dependent LTP at their excitatory synapses after repeated activation of the EC input. Using optogenetic stimulation and a transgenic Ndnf-Cre mouse line, they showed that this LTP is formed in neurogliaform cells at the temporoammonic (i.e. EC) input synapses, but not at the thalamic input synapses. Finally, theta-burst stimulation of the EC input induced LTP at the synapses of SLM neurogliaform cells, but not at the temporoammonic pathway to pyramidal cell synapses.

Electrogenic phenomena in distal dendrites of principal neurons in the hippocampus have a major role in gating signaling and synaptic plasticity at afferent synapses on proximal dendrites. Therefore, understanding how the recruitment and LTP formation at the EC to neurogliaform cells contribute to excitation-inhibition balance at the temporoammonic-pyramidal cell synapses is an important topic. Overall, this is an interesting and important study. However, there are some major and minor issues that need to be addressed.

Major issues:

1. The authors showed that LFS-P (200 pulses at 2 Hz) electrical stimulation paired with postsynaptic depolarization at -10 mV in voltage clamp induced NMDAR-dependent LTP in interneurons in mouse SLM. The authors stated that such plasticity can be induced by selective optogenetic stimulation of afferents in the temporoammonic pathway from the entorhinal cortex,

but not by equivalent stimulation of afferents from the thalamic nucleus reuniens. This conclusion is incorrect. Although the frequency of optogenetic stimulation (200 pulses at 2 Hz) is identical to the electrical stimulation, the mechanism underlying optogenetic LTP may differ from electrical stimulation at afferents. During each optogenetic stimulation, additional Ca²⁺ influx through channelrhodopsin channels at presynaptic terminals may lead to induction of different forms of LTP. The authors should perform additional experiments to verify whether optogenetic LFS-P-induced LTP is indeed NMDAR-dependent.

2. An interesting finding in this study is that optogenetic LFS-P induced LTP at the EC input but not thalamic input. The readers may get confused by this conclusion because this idea may not be valid for other stimulation protocols. In this study, the authors demonstrated that four different induction protocols (electrical and optogenetic LFS-P, STDP, TBS) can evoke LTP in SLM interneurons at the EC input. It remains unclear that optogenetic LFS-P, STDP and TBS cannot induce LTP in SLM interneurons at the thalamic input.

3. The comparison of the EPSPs at the EC and thalamic inputs (Figure 4I and J) is not compelling. The optogenetic-evoked response at each input depends on virus expression efficiency and the light intensity. These two inputs may exhibit different light intensity and EPSP (input-output) relationships. Did the authors use the maximal light intensity to evoke EPSPs? Do these two inputs display different input-output relationships?

4. In this study, the authors applied different LTP induction protocols under different conditions. However, each protocol may induce LTP with different underlying mechanisms. In the Figure 7A, the authors showed that electrical TBS evoked LTP in the presence of GABA receptor blockers. In the Figure 7E, optogenetic EC-TBS induced LTP of disynaptic IPSCs (black), but not of monosynaptic EPSCs (white). If the authors want to demonstrate that intact GABAergic inhibition can prevent LTP of monosynaptic EPSCs, they should demonstrate that optogenetic EC-TBS (rather than electrical TBS in figure 7A) can evoke LTP in the presence of GABA blockers.

Minor issues:

1. The authors claimed run-up at the thalamic input by optogenetic stimulation (Figure 4E n= 3), the sample size is not sufficient to make the conclusion. The error bar denotes SEM in this study (the authors should change to SD according to the journal's policy). The error bars (SEM) are large. For these three recordings, did the author ensure the stability of serial resistance of these "3" cells? The authors need to increase the recording number to increase the statistic power before making any conclusion.

2. The authors did not explain why two stimulation electrodes were needed in the figure 1.

3. For the figure 2A, relative to LFS-P group, the authors did not describe how the control was conducted.

4. According to the location of these two stimulation electrodes (S1 and S2), these two stimulation electrodes should stimulate largely overlapped population of axons in the SLM. How can they separate the stimulated populations by different electrodes (S1, S2)?

5. For the whole cell recording of EPSP, the authors should explain why they measured the initial slope of EPSP instead of the peak amplitude.

6. In figure 1B, authors performed a random patch of 19 SLM interneurons and stated on page 12, line 16 that only some of these cells were morphologically identified as NGF-like cells. How many of these cells were identified as NGF-like cells? What are the electrophysiological and morphological features of the non-NGF or morphologically unidentified cells?

7. To examine the pathway-specific potentiation and the spread of stimulation (Figure 2A), the authors should show the data to confirm S1 and S2 inputs are independent

8. The sample number in figure 2A is n=6. The authors concluded that LFS-P induced LTP was largely pathway-specific. However, the P value =0.08. It is unclear why the authors can conclude pathway-specific LTP here. Since there is a trend for statistical significance (P value 0.05), the authors should increase the sample size in Fig. 2A.

9. Spell out STDP (Figure 3).

10. Optogenetic EC activation induced di-synaptic IPSCs, which can be largely blocked by NBQX. However, the electrical stimulation in the SLM should evoke both mono-synaptic and di-synaptic IPSCs in CA1 pyramidal cells (Figure 7C). It is unclear why NBQX had no effect on IPSCs. The authors should explain this.

11. In the figure 7C, the mono-synaptic IPSCs (stim IPSCs) should exhibit a shorter latency than optogenetically evoked disynaptic IPSCs. However, the traces of light evoked-IPSCs did not show longer latency. Did the authors illustrate the light stimulation (blue tick) correctly? Please analyze the latency of stim and light groups in Fig. 7C. In figure 1, given that some cells underwent LTD, the authors should confirm the identity of these cells whether they are NGF or non-NGF cells.

12. In the methods, the authors explained that stimulation electrode S1 and S2 activated independent pathways. In fact, it is difficult to selectively stimulate two independent pathways using electrical stimulation. If they can demonstrate this, they

should provide data as they claimed in the methods. This data should be incorporated to figure 1.

13. The bar graph in figure 3B is confusing as it shows the comparison within groups. The author should compare the LTP of STDP and STDP+APV.

14. In figure 4D and E, the authors monitored the stability of light-evoked EPSPs at the EC and thalamic input synapses. However, the traces were represented as "pre" and "post" like in LTP experiments. Using the same terms can confuse readers.

Referee #2:

The paper by Mercier and co-workers is aimed at characterizing the role of long-term potentiation (LTP) of glutamatergic synapses in neurogliaform (NF) interneurons located in the stratum lacunosum moleculare (LM) on the excitatory-inhibitory balance in the temporoammonic pathway. The authors show that LTP induced by LFS-P but not by STDP protocol depends on NMDA receptors. Interneurons located in the LM are known to receive two types of inputs: from the entorhinal cortex (EC) and from the thalamus. Using optogenetic tools, they show that only EC inputs and not those from the thalamus express LTP induced by theta bursts. Finally, the authors show that an increase in di-synaptic inhibition but not in monosynaptic excitation is observed following theta burst stimulation of EC. This results indicates that LTP in NF interneurons promotes inhibition over excitation.

The paper is interesting and provide novel findings but the study needs additional experiments to confirm the results.

Specific points:

1. Statistics: t-test cannot be used on small samples as $n = 5$ or 7 . Please use non-parametric tests in all statistics (Mann-Whitney and Wilcoxon tests).
2. Figure 2A: the lack of input specificity in LTP induced by LFS-P is surprising and somehow questionable (paired input: +35% control input: +15%, not significantly different). As the independence of their two inputs has been verified with paired-pulse paradigm, what could be the reason for this? In addition, I suggest to increase the n to 10 to know whether the observed trend goes towards a true difference or not.
3. Figure 3: given the previous results, the STDP-induced LTP that does not require NMDA receptor is puzzling. What is the fundamental difference between the two protocols (i.e. LFS-P and STDP-LTP)? As STDP-LTP requires Ca^{2+} influx mediated by voltage-gated calcium channels, the authors should have tested spiking alone.
4. Figure 4E. The number of neurons recorded with optogenetic stimulation of the thalamus is too low ($n = 3$). Please add at least two more cells to confirm the run-up and thus, to solidify the study.
5. What is the value of liquid junction potential? This value should be indicated in the manuscript.

Minor points:

1. Title: please use the same term in the short and long title to define interneurons (either neurogliaforms interneurons or lacunosum moleculare interneurons).
2. Key summary: theta-burst activity is used only in part of Fig. 6. Is that really the main finding of the manuscript?
3. Page 4, please define NRe.

END OF COMMENTS

Confidential Review

16-Dec-2021

Dear Professor Kullmann,

Re: JP-RP-2021-282753 "Long-term potentiation in stratum lacunosum-moleculare interneurons modulates excitation-inhibition balance in the temporoammonic pathway" by Marion Mercier, Vincent Magloire, Jonathan Cornford, and Dimitri M Kullmann

Thank you for submitting your manuscript to The Journal of Physiology. It has been assessed by a Reviewing Editor and by 2 expert Referees and I am pleased to tell you that it is considered to be acceptable for publication following satisfactory revision.

The reports are copied at the end of this email. Please address all of the points and incorporate all requested revisions, or explain in your Response to Referees why a change has not been made.

NEW POLICY: In order to improve the transparency of its peer review process The Journal of Physiology publishes online as supporting information the peer review history of all articles accepted for publication. Readers will have access to decision letters, including all Editors' comments and referee reports, for each version of the manuscript and any author responses to peer review comments. Referees can decide whether or not they wish to be named on the peer review history document.

Authors are asked to use The Journal's premium BioRender (<https://biorender.com/>) account to create/redrawn their Abstract Figures. Information on how to access The Journal's premium BioRender account is here: <https://physoc.onlinelibrary.wiley.com/journal/14697793/biorender-access> and authors are expected to use this service. This will enable Authors to download high-resolution versions of their figures.

I hope you will find the comments helpful and have no difficulty returning your revisions within 4 weeks.

Your revised manuscript should be submitted online using the links in Author Tasks <https://jp.msubmit.net/cgi-bin/main.plex?el=A2JS6Etv6A3IsF5F7A9ftdOR2sXQDi4ki9y7N9dICNQZ>.

Any image files uploaded with the previous version are retained on the system. Please ensure you replace or remove all files that have been revised.

REVISION CHECKLIST:

- Article file, including any tables and figure legends, must be in an editable format (eg Word)
- Abstract figure file (see above)
- Statistical Summary Document
- Upload each figure as a separate high quality file
- Upload a full Response to Referees, including a response to any Senior and Reviewing Editor Comments;

- Upload a copy of the manuscript with the changes highlighted.

- A potential 'Cover Art' file for consideration as the Issue's cover image;

- Appropriate Supporting Information (Video, audio or data set https://jp.msubmit.net/cgi-bin/main.plex?form_type=display_requirements#supp).

To create your 'Response to Referees' copy all the reports, including any comments from the Senior and Reviewing Editors, into a Word, or similar, file and respond to each point in colour or CAPITALS and upload this when you submit your revision.

I look forward to receiving your revised submission.

If you have any queries please reply to this email and staff will be happy to assist.

Yours sincerely,

David Wyllie
Senior Editor
The Journal of Physiology

REQUIRED ITEMS:

-Author photo and profile. First (or joint first) authors are asked to provide a short biography (no more than 100 words for one author or 150 words in total for joint first authors) and a portrait photograph. These should be uploaded and clearly labelled with the revised version of the manuscript. See Information for Authors for further details.

-You must start the Methods section with a paragraph headed Ethical Approval. A detailed explanation of journal policy and regulations on animal experimentation is given in Principles and standards for reporting animal experiments in The Journal of Physiology and Experimental Physiology by David Grundy J Physiol, 593: 2547-2549. doi:10.1113/JP270818.). A checklist outlining these requirements and detailing the information that must be provided in the paper can be found at: <https://physoc.onlinelibrary.wiley.com/hub/animal-experiments>. Authors should confirm in their Methods section that their experiments were carried out according to the guidelines laid down by their institution's animal welfare committee, and conform to the principles and regulations as described in the Editorial by Grundy (2015). The Methods section must contain details of the anaesthetic regime: anaesthetic used, dose and route of administration and method of killing the experimental animals.

-Your manuscript must include a complete Additional Information section

-Please upload separate high-quality figure files via the submission form.

-A Statistical Summary Document, summarising the statistics presented in the manuscript, is required upon revision. It must be on the Journal's template, which can be downloaded from the link in the Statistical Summary Document section here: https://jp.msubmit.net/cgi-bin/main.plex?form_type=display_requirements#statistics

-Papers must comply with the Statistics Policy https://jp.msubmit.net/cgi-bin/main.plex?form_type=display_requirements#statistics

In summary:

-If $n \leq 30$, all data points must be plotted in the figure in a way that reveals their range and distribution. A bar graph with data points overlaid, a box and whisker plot or a violin plot (preferably with data points included) are acceptable formats.

-If $n > 30$, then the entire raw dataset must be made available either as supporting information, or hosted on a not-for-profit repository e.g. FigShare, with access details provided in the manuscript.

-'n' clearly defined (e.g. x cells from y slices in z animals) in the Methods. Authors should be mindful of pseudoreplication.

-All relevant 'n' values must be clearly stated in the main text, figures and tables, and the Statistical Summary Document (required upon revision)

-The most appropriate summary statistic (e.g. mean or median and standard deviation) must be used. Standard Error of the Mean (SEM) alone is not permitted.

-Exact p values must be stated. Authors must not use 'greater than' or 'less than'. Exact p values must be stated to three significant figures even when 'no statistical significance' is claimed.

-Statistics Summary Document completed appropriately upon revision

-Please include an Abstract Figure. The Abstract Figure is a piece of artwork designed to give readers an immediate understanding of the research and should summarise the main conclusions. If possible, the image should be easily 'readable' from left to right or top to bottom. It should show the physiological relevance of the manuscript so readers can assess the importance and content of its findings. Abstract Figures should not merely recapitulate other figures in the manuscript. Please try to keep the diagram as simple as possible and without superfluous information that may distract from the main conclusion(s). Abstract Figures must be provided by authors no later than the revised manuscript stage and should be uploaded as a separate file during online submission labelled as File Type 'Abstract Figure'. Please ensure that you include the figure legend in the main article file. All

Abstract Figures should be created using BioRender. Authors should use The Journal's premium BioRender account to export high-resolution images. Details on how to use and access the premium account are included as part of this email.

EDITOR COMMENTS

Reviewing Editor:

Both Reviewers agree that these findings are exciting.

They have further suggestions for additional experiments and points for clarification (e.g. is the ability of certain synapses to undergo plasticity dependent on the properties synapses or method of activation (electrical vs optical)).

In addition they suggest an increase in robustness of results is needed by increasing in 'n' for certain experiments. This might also affect the statistical analysis that is most appropriate (non-parametric or t-tests).

Senior Editor:

Your manuscript has been positively assessed by two expert referees and an Reviewing Editor. Notwithstanding their overall positive comments, they have raised concerns about some aspects of the work which will require further experimentation. Aside from the specific queries, they note that some experiments would appear to be under-powered and I would encourage you to ensure that that data you report are statistically robust and analysed appropriately. Please also ensure you report n values clearly - I note that in places you give x slices from y animals but in other places you simply report the number of cells (from how many slices, how many animals?). I look forward to receiving a revised manuscript in due course.

Please ensure that any revised manuscript complies with the Statistics Policy - for example, report SD (not SEM), precise P values (unless $P < 0.0001$)

We thank you for considering our manuscript for publication in the Journal of Physiology. We have addressed and responded to the reviewers' comments to the best of our ability, as detailed below.

We have increased n values for experiments that were under-powered. With regards to reporting of n values, we include a sentence at the start of the Statistical Analysis section, detailing that for all LTP experiments (the majority of the paper) n values represent separate slices from at least 3 different animals. We specifically report the number of animals for quantification of expression in Ndnf-Cre animals (Figure 5) where significant between-animal variability can occur due to viral injections.

REFeree COMMENTS

Referee #1:

The authors of this manuscript investigated long-term potentiation (LTP) in interneurons in the stratum lacunosum-moleculare (SLM) and how it increases di-synaptic inhibition of pyramidal cells, thus finally shifting excitation-inhibition balance at the entorhinal cortex (EC) input to pyramidal cell synapses. They demonstrated that SLM interneurons, which were mostly identified as neurogliaform cells, expressed Hebbian NMDA receptor-dependent LTP at their excitatory synapses after repeated activation of the EC input. Using optogenetic stimulation and a transgenic *Ndnf-Cre* mouse line, they showed that this LTP is formed in neurogliaform cells at the temporoammonic (i.e. EC) input synapses, but not at the thalamic input synapses. Finally, theta-burst stimulation of the EC input induced LTP at the synapses of SLM neurogliaform cells, but not at the temporoammonic pathway to pyramidal cell synapses.

Electrogenic phenomena in distal dendrites of principal neurons in the hippocampus have a major role in gating signaling and synaptic plasticity at afferent synapses on proximal dendrites. Therefore, understanding how the recruitment and LTP formation at the EC to neurogliaform cells contribute to excitation-inhibition balance at the temporoammonic-pyramidal cell synapses is an important topic. Overall, this is an interesting and important study. However, there are some major and minor issues that need to be addressed.

We thank the reviewer for their positive assessment of the study and insightful comments.

Major issues:

1. The authors showed that LFS-P (200 pulses at 2 Hz) electrical stimulation paired with postsynaptic depolarization at -10 mV in voltage clamp induced NMDAR-dependent LTP in interneurons in mouse SLM. The authors stated that such plasticity can be induced by selective optogenetic stimulation of afferents in the temporoammonic pathway from the entorhinal cortex, but not by equivalent stimulation of afferents from the thalamic nucleus reuniens. This conclusion is incorrect. Although the frequency of optogenetic stimulation (200 pulses at 2 Hz) is identical to the electrical stimulation, the mechanism underlying optogenetic LTP may differ from electrical stimulation at afferents. During each optogenetic stimulation, additional Ca²⁺ influx through channelrhodopsin channels at presynaptic terminals may lead to induction of different forms of LTP. The authors should perform additional experiments to verify whether optogenetic LFS-P-induced LTP is indeed NMDAR-dependent.

We agree that optogenetically-induced LTP may differ from electrically-induced LTP. However, the question raised in Figure 4 concerns whether LTP can be induced at afferents from the EC, the thalamus, or both, and we feel that our experiments allow us to confidently conclude that LTP (albeit optogenetically-induced) occurs at EC inputs only. This conclusion is further strengthened by the additional experiment in panel 4J, performed at the request of both reviewers.

Furthermore, if LTP were due to a presynaptic mechanism, PPR would be expected to change. PPR data was obtained in $n = 10$ EC-LTP experiments and shows no significant change in PPR (baseline PPR: $195 \pm 60\%$, post-LTP PPR: $180 \pm 44\%$, $t(9) = 2.08$, $p = 0.07$). While there was a small non-significant trend towards a decrease in PPR, it did not correlate with the magnitude of LTP ($p = 0.41$, $p = 0.244$, Spearman correlation).

2. An interesting finding in this study is that optogenetic LFS-P induced LTP at the EC input but not thalamic input. The readers may get confused by this conclusion because this idea may not be valid

for other stimulation protocols. In this study, the authors demonstrated that four different induction protocols (electrical and optogenetic LFS-P, STDP, TBS) can evoke LTP in SLM interneurons at the EC input. It remains unclear that optogenetic LFS-P, STDP and TBS cannot induce LTP in SLM interneurons at the thalamic input.

We agree that it is important to test other stimulation protocols to assess LTP at thalamic inputs more comprehensively. Because STDP induces a form of plasticity that is at least in part due to post-synaptic back-propagating action potentials (see Reviewer 2 point 3, and new figure panel 3D), thereby diminishing the influence of the specific pre-synaptic input, we chose to use a TBS protocol, in line with that used in Figures 6 and 7. We found no LTP following thalamic TBS, strengthening our conclusion.

3. The comparison of the EPSPs at the EC and thalamic inputs (Figure 4I and J) is not compelling. The optogenetic-evoked response at each input depends on virus expression efficiency and the light intensity. These two inputs may exhibit different light intensity and EPSP (input-output) relationships. Did the authors used the maximal light intensity to evoke EPSPs? Do these two inputs display different input-output relationships?

This is an important point. The potential difference in input-output between animals/slices/inputs is precisely why we chose not to use a set light intensity, but rather aim for postsynaptic responses that were roughly of the same magnitude, and importantly were submaximal (we have added this detail in the methods). Optogenetic responses were quite variable so the overall range of response sizes was fairly broad, but importantly was not different between the two groups, demonstrating that the difference in LTP could not be attributed to differences in the amount of optogenetically-induced postsynaptic depolarisation.

4. In this study, the authors applied different LTP induction protocols under different conditions. However, each protocol may induce LTP with different underlying mechanisms. In the Figure 7A, the authors showed that electrical TBS evoked LTP in the presence of GABA receptor blockers. In the Figure 7E, optogenetic EC-TBS induced LTP of disynaptic IPSCs (black), but not of monosynaptic EPSCs (white). If the authors want to demonstrate that intact GABAergic inhibition can prevent LTP of monosynaptic EPSCs, they should demonstrate that optogenetic EC-TBS (rather than electrical TBS in figure 7A) can evoke LTP in the presence of GABA blockers.

We agree that Fig. 7A is a distraction. The purpose of the experiments illustrated in Fig. 7 was to assess how LTP at EC-SLM interneurons impacts on disynaptic inhibition of pyramidal cells. The conclusion that there was no potentiation of EPSCs in pyramidal cells with inhibition intact is hardly surprising given the powerful feed-forward inhibition experienced by distal dendrites. The important finding is that disynaptic inhibition was, by contrast, potentiated by EC-TBS. We have therefore removed Fig. 7A for clarity.

Minor issues:

1. The authors claimed run-up at the thalamic input by optogenetic stimulation (Figure 4E n= 3), the sample size is not sufficient to make the conclusion. The error bar denotes SEM in this study (the authors should change to SD according to the journal's policy). The error bars (SEM) are large. For these three recordings, did the author ensure the stability of serial resistance of these "3" cells? The authors need to increase the recording number to increase the statistic power before making any conclusion.

We agree with the reviewer and have carried out additional experiments. The dataset is now $n = 6$, and large run-up was seen in 4/6 experiments. While this increase was not significant overall ($p = 0.077$), it is enough to confound the interpretation of any experiments performed using optogenetic stimulation of thalamic fibers to probe for LTP.

Series resistance was verified in all experiments and any with a change $> 20\%$ were discarded, as described in the methods. We have changed all summary bar graphs to mean \pm SD.

2. The authors did not explain why two stimulation electrodes were needed in the figure 1.

This is for the dual pathway experiment in Fig. 2. We have added a reference to the diagram in Fig. 1A at the start of the paragraph on the dual pathway experiment. This is also described in the methods.

3. For the figure 2A, relative to LFS-P group, the authors did not describe how the control was conducted.

We have added a sentence about this in the methods section.

4. According to the location of these two stimulation electrodes (S1 and S2), these two stimulation electrodes should stimulate largely overlapped population of axons in the SLM. How can they separate the stimulated populations by different electrodes (S1, S2)?

Electrodes were placed as far away from each other as possible at opposite ends of SLM and pathway independence was verified by comparing paired-pulse ratio within and between electrodes, as described in the methods section. We have added a panel in Figure 2 (B) depicting this and added some discussion of this in the results section.

5. For the whole cell recording of EPSP, the authors should explain why they measured the initial slope of EPSP instead of the peak amplitude.

Initial slope was selected to ensure measurement of monosynaptic inputs, and to prevent any contamination of response measurement by spikelets which sometimes occurred on top of larger EPSPs. Amplitude was measured when looking at disynaptic responses.

6. In figure 1B, authors performed a random patch of 19 SLM interneurons and stated on page 12, line 16 that only some of these cells were morphologically identified as NGF-like cells. How many of these cells were identified as NGF-like cells? What are the electrophysiological and morphological features of the non-NGF or morphologically unidentified cells?

We were unfortunately unable to reconstruct sufficient cells to draw conclusions based on morphology. We have however explored cell variability and how this relates to LTP further by analysing electrophysiological properties, and find that a number of properties correlate with LTP magnitude. This has been added to Figure 1 (C-F) and is discussed in the results section.

7. To examine the pathway-specific potentiation and the spread of stimulation (Figure 2A), the authors should show the data to confirm S1 and S2 inputs are independent

This has been added to Figure 2, panel B.

8. The sample number in figure 2A is $n=6$. The authors concluded that LFS-P induced LTP was largely

pathway-specific. However, the P value =0.08. It is unclear why the authors can conclude pathway-specific LTP here. Since there is a trend for statistical significance (P value 0.05), the authors should increase the sample size in Fig. 2A.

We agree and have increased the sample size to n = 10, and now see a significant difference between the two pathways. We have changed the discussion in the results section to reflect this.

9. Spell out STDP (Figure 3).

We have changed this to “spike pairing” in line with the terminology used in the manuscript.

10. Optogenetic EC activation induced di-synaptic IPSCs, which can be largely blocked by NBQX. However, the electrical stimulation in the SLM should evoke both mono-synaptic and di-synaptic IPSCs in CA1 pyramidal cells (Figure 7C). It is unclear why NBQX had no effect on IPSCs. The authors should explain this.

Indeed this was at first sight unexpected. However, it is consistent with a high density of SLM interneurons with profuse axonal arborizations, such that electrical stimulation recruits powerful monosynaptic inhibition, which is much greater than the disynaptic component. We have added a sentence to highlight this in the text.

11. In the figure 7C, the mono-synaptic IPSCs (stim IPSCs) should exhibit a shorter latency than optogenetically evoked disynaptic IPSCs. However, the traces of light evoked-IPSCs did not show longer latency. Did the authors illustrate the light stimulation (blue tick) correctly? Please analyze the latency of stim and light groups in Fig. 7C.

We agree and have analysed latency of responses. Unfortunately, responses to stimulation were often too contaminated by the stimulation artifact to give an accurate measure - a reflection of the short response latency, as would be expected for predominantly direct excitation. Responses to optogenetic stimulation however had a mean latency of 7.03 ± 2 (SD) ms, which is in line with slow dendritic disynaptic inhibition (Pouille & Scanziani, 2001). We have changed the traces in Fig. 7C for optogenetic responses with a more representative delay, although we appreciate this is hard to see on such a small scale.

In figure 1, given that some cells underwent LTD, the authors should confirm the identity of these cells whether they are NGF or non-NGF cells.

We agree that this would be interesting to know but are unfortunately unable to conclusively determine which cells were NGF or non-NGF. However, we have attempted to address this at least partially by exploring electrophysiological properties and how these relate to LTP, and find that cells with properties that are in line with those of NGF cells tended to undergo larger LTP (Fig. 1C-F). This would suggest that those expressing LTD were not NGF cells.

12. In the methods, the authors explained that stimulation electrode S1 and S2 activated independent pathways. In fact, it is difficult to selectively stimulate two independent pathways using electrical stimulation. If they can demonstrate this, they should provide data as they claimed in the methods. This data should be incorporated to figure 1.

This has been added to Figure 2, panel B, and the possibility of contamination between the two pathways is discussed in the results section.

13. The bar graph in figure 3B is confusing as it shows the comparison within groups. The author should compare the LTP of STDP and STDP+APV.

We have added a between groups comparison to the figure and added the statistics to the results section.

14. In figure 4D and E, the authors monitored the stability of light-evoked EPSPs at the EC and thalamic input synapses. However, the traces were represented as "pre" and "post" like in LTP experiments. Using the same terms can confuse readers.

We agree and have changed this to "start" and "end".

Referee #2:

The paper by Mercier and co-workers is aimed at characterizing the role of long-term potentiation (LTP) of glutamatergic synapses in neurogliaform (NF) interneurons located in the stratum lacunosum moleculare (LM) on the excitatory-inhibitory balance in the temporoammonic pathway. The authors show that LTP induced by LFS-P but not by STDP protocol depends on NMDA receptors. Interneurons located in the LM are known to receive two types of inputs: from the entorhinal cortex (EC) and from the thalamus. Using optogenetic tools, they show that only EC inputs and not those from the thalamus express LTP induced by theta bursts. Finally, the authors show that an increase in di-synaptic inhibition but not in monosynaptic excitation is observed following theta burst stimulation of EC. This results indicates that LTP in NF interneurons promotes inhibition over excitation.

The paper is interesting and provide novel findings but the study needs additional experiments to confirm the results.

We thank the reviewer for their positive comments and valuable suggestions.

Specific points:

1. Statistics: t-test cannot be used on small samples as $n = 5$ or 7 . Please use non-parametric tests in all statistics (Mann-Whitney and Wilcoxon tests).

We agree with the reviewer in the sense that the t -test assumes the distribution of sample means will be normally distributed. However, while means of larger (20-30) samples from population distribution with finite variance approach normal by the central limit theorem regardless of the underlying population distribution, this does not mean that the means of smaller samples will not be normal. Instead, the number of samples required is a function of the non-normality of the population distribution. Indeed, if the population distribution is normal, the distribution of means will also always be normal. Therefore, we do not agree with the reviewer's statement that t -tests cannot be used on small samples.

In addition, we would also like to highlight that the t -test is robust to small samples of non-normal data. This can be verified with simulations, justifying the continued use of t -tests in the literature with sample sizes of $n = 5-7$.

2. Figure 2A: the lack of input specificity in LTP induced by LFS-P is surprising and somehow

questionable (paired input: +35% control input: +15%, not significantly different). As the independence of their two inputs has been verified with paired-pulse paradigm, what could be the reason for this? In addition, I suggest to increase the n to 10 to know whether the observed trend goes towards a true difference or not.

We agree and have increased the sample size to n =10 as suggested. The difference between pathways is now significant, but a small potentiation is also seen in the control pathway which we discuss in the results section.

3. Figure 3: given the previous results, the STDP-induced LTP that does not require NMDA receptor is puzzling. What is the fundamental difference between the two protocols (i.e. LTS-P and STDP-LTP)? As STDP-LTP requires Ca²⁺ influx mediated by voltage-gated calcium channels, the authors should have tested spiking alone.

We agree that this is surprising and have added a dataset testing spiking alone, as suggested (Fig. 3D). This revealed a smaller potentiation that did not quite reach statistical significance, suggesting that this form of LTP requires depolarisation, and therefore Ca²⁺ influx through VGCCs, mediated by both back-propagating action potentials and dendritic EPSPs.

4. Figure 4E. The number of neurons recorded with optogenetic stimulation of the thalamus is too low (n = 3). Please add at least two more cells to confirm the run-up and thus, to solidify the study.

We agree and have increased the sample size to n = 6.

5. What is the value of liquid junction potential? This value should be indicated in the manuscript.

We have added this to the methods section.

Minor points:

1. Title: please use the same term in the short and long title to define interneurons (either neurogliaforms interneurons or lacunosum moleculare interneurons).

We thank the reviewer for highlighting this and have changed the title.

2. Key summary: theta-burst activity is used only in part of Fig. 6. Is that really the main finding of the manuscript?

TBS is the most physiologically relevant protocol used and is also used in Figure 7 where the overall function of the plasticity on the hippocampal network is demonstrated.

3. Page 4, please define NRe.

This is defined on page 3.

END OF COMMENTS

The Physiological Society is a company limited by guarantee. Registered in England and Wales, No. 00323575. Registered Office: Hodgkin Huxley House, 30 Farringdon Lane, London, EC1R 3AW, UK. Registered Charity No. 211585. The Physiological Society and The Journal of Physiology are registered trademarks.

This email and any files transmitted with it are confidential and intended solely for the use of the individual or entity to whom they are addressed. If you have received this email in error please notify the sender. If you are not the named addressee you should not disseminate, distribute or copy this e-mail. The Physiological Society may monitor email traffic data.

The Physiological Society has taken reasonable precautions to ensure no viruses are present in this email, however does not accept responsibility for any loss or damage arising from the use of this email or attachments.

Dear Dr Kullmann,

Re: JP-RP-2022-282753R1 "Long-term potentiation in neurogliaform interneurons modulates excitation-inhibition balance in the temporoammonic pathway" by Marion Mercier, Vincent Magloire, Jonathan Cornford, and Dimitri M Kullmann

I am pleased to tell you that your paper has been accepted for publication in The Journal of Physiology.

NEW POLICY: In order to improve the transparency of its peer review process The Journal of Physiology publishes online as supporting information the peer review history of all articles accepted for publication. Readers will have access to decision letters, including all Editors' comments and referee reports, for each version of the manuscript and any author responses to peer review comments. Referees can decide whether or not they wish to be named on the peer review history document.

The last Word version of the paper submitted will be used by the Production Editors to prepare your proof. When this is ready you will receive an email containing a link to Wiley's Online Proofing System. The proof should be checked and corrected as quickly as possible.

Authors should note that it is too late at this point to offer corrections prior to proofing. The accepted version will be published online, ahead of the copy edited and typeset version being made available. Major corrections at proof stage, such as changes to figures, will be referred to the Reviewing Editor for approval before they can be incorporated. Only minor changes, such as to style and consistency, should be made a proof stage. Changes that need to be made after proof stage will usually require a formal correction notice.

All queries at proof stage should be sent to TJP@wiley.com

Are you on Twitter? Once your paper is online, why not share your achievement with your followers. Please tag The Journal (@jphysiol) in any tweets and we will share your accepted paper with our 23,000+ followers!

Yours sincerely,

David Wyllie
Senior Editor
The Journal of Physiology

P.S. - You can help your research get the attention it deserves! Check out Wiley's free Promotion Guide for best-practice recommendations for promoting your work at www.wileyauthors.com/eeo/guide. And learn more about Wiley Editing Services which offers professional video, design, and writing services to create shareable video abstracts, infographics, conference posters, lay summaries, and research news stories for your research at www.wileyauthors.com/eeo/promotion.

*** IMPORTANT NOTICE ABOUT OPEN ACCESS ***

To assist authors whose funding agencies mandate public access to published research findings sooner than 12 months after publication The Journal of Physiology allows authors to pay an open access (OA) fee to have their papers made freely available immediately on publication.

You will receive an email from Wiley with details on how to register or log-in to Wiley Authors Services where you will be able to place an OnlineOpen order.

You can check if your funder or institution has a Wiley Open Access Account here <https://authorservices.wiley.com/author-resources/Journal-Authors/licensing-and-open-access/open-access/author-compliance-tool.html>

Your article will be made Open Access upon publication, or as soon as payment is received.

If you wish to put your paper on an OA website such as PMC or UKPMC or your institutional repository within 12 months of publication you must pay the open access fee, which covers the cost of publication.

OnlineOpen articles are deposited in PubMed Central (PMC) and PMC mirror sites. Authors of OnlineOpen articles are permitted to post the final, published PDF of their article on a website, institutional repository, or other free public server, immediately on publication.

Note to NIH-funded authors: The Journal of Physiology is published on PMC 12 months after publication, NIH-funded authors DO NOT NEED to pay to publish and DO NOT NEED to post their accepted papers on PMC.

EDITOR COMMENTS

Reviewing Editor:

No further comments.

Senior Editor:

Thank you for revising your manuscript in accordance with the reviewers' comments and in particular for performing the necessary additional experiments. We have consulted with our Statistics Editor regarding the use of t-tests and they agree that it can be used on the small sample sizes that you have in some datasets, so in this respect I am happy to accept your manuscript for publication.

REFEREE COMMENTS

Referee #1:

The authors have properly addressed my concerns by adding new experiments and editing the text.

Referee #2:

Most concerns have been addressed. I still do not agree with the use of t-test. This test is not discriminant for small samples.

Referee #3:

One of the reviewers maintains that the t test cannot be used in small samples. However, sample sizes of four were used as an example of the method in Student's 1908 paper on the Probable Error of a Mean and, indeed, a robust small-sample method was Student's aim, rather than relying on the large-sample Z theory in use throughout the 19th Century. Assessing the assumptions underpinning the test is not robust with very small sample sizes but, as the authors note, the t test is remarkably robust to departures from normality. Simulations using resampling from uniform distributions (about as non-normal as it gets) show that the t test is robust; it is much more susceptible to non-constant error variance (heteroscedasticity) than to non-normality. More critical is the likely low-power in this study for detecting small but physiologically meaningful effects. In noisy, low-power settings, results that are observed to be statistically significant are likely (grossly) exaggerated versus their true value; known as Type M (magnitude) error. Caution is warranted in trying to make any definitive inferences from small studies. But I have no real issue with using a t test on a small sample.